# OTTER: Effortless Label Distribution Adaptation of Zero-shot Models

**Changho Shin, Jitian Zhao, Sonia Cromp, Harit Vishwakarma, Frederic Sala**

Department of Computer Sciences
University of Wisconsin-Madison
{cshin23, jzhao326, cromp, hvishwakarma, fsala}@wisc.edu

## Abstract

Popular zero-shot models suffer due to artifacts inherited from pretraining. One particularly detrimental issue, caused by unbalanced web-scale pretraining data, is *mismatched label distribution*. Existing approaches that seek to repair the label distribution are not suitable in zero-shot settings, as they have mismatching requirements, such as needing access to labeled downstream task data or knowledge of the true label balance in the pretraining distribution. We sidestep these challenges and introduce a simple and lightweight approach to adjust pretrained model predictions via optimal transport. Our technique requires only an *estimate* of the label distribution of a downstream task. Theoretically, we characterize the improvement produced by our procedure under certain mild conditions and provide bounds on the error caused by misspecification. Empirically, we validate our method in a wide array of zero-shot image and text classification tasks, improving accuracy by 4.8% and 15.9% on average, and beating baselines like prior matching—often by significant margins—in 17 out of 21 datasets.

## 1 Introduction

Zero-shot models are popular but struggle with biases inherited from their large pretraining datasets [21; 59; 2]. In particular, zero-shot classification is strongly biased by *the label distribution* of the pretraining task. When the label distribution of the downstream task differs from pretraining, the performance of zero-shot classifiers suffers greatly. For example, Figure 1 illustrates the effects of mismatched distributions on a pet image classification task. Two CLIP models (RN50, and ViT-B/16) produce biased predictions on the `Abyssinian` and `Persian` classes. Furthermore, datasets with a large number of classes, such as ImageNet, may contain both extremely common and very rare classes, resulting in an outsized probability that a zero-shot model will predict some classes over others. As a result, even large models intended for use in zero-shot settings, such as CLIP [49], naturally have a label distribution mismatch between pretraining data and downstream tasks.

Existing methods that seek to address label distribution *make strong assumptions or have expensive requirements*. For example, to fine-tune a model, we must obtain a labeled fine-tuning dataset of adequate size, then obtain the time and compute to further train the model. To perform label shift adap-

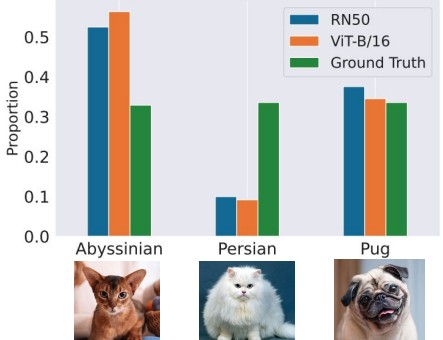
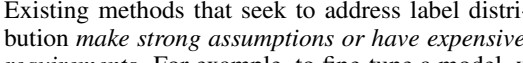

Figure 1: Label distribution mismatch example in zero-shot classification. In the Oxford-IIIT-Pet dataset, the ground-truth labels are uniformly distributed, while zero-shot models exhibit biased predictions toward certain classes. This bias is influenced by the distribution of labels in the pretraining task.

38th Conference on Neural Information Processing Systems (NeurIPS 2024).

tation techniques, we must know the true label distribution of the pretraining distribution—difficult to impossible for real-world tasks.

Can we deal with label distribution mismatch ***without additional training or access to ground-truth downstream task information***? While seemingly challenging, one cause for optimism is the observation that zero-shot models still give relatively high prediction probabilities for correct classes, though classes common in pretraining tend to have relatively inflated scores overall. Intuitively, the model has already learned to identify examples of its downstream classes (and so does not require further training) and is already impacted by the pretraining label distribution (and so does not need access to the ground-truth pretraining label distribution). Instead, the model's prediction probabilities must be adjusted based on an estimated downstream label distribution specification.

To perform this label distribution adjustment, we view zero-shot learning through the lens of optimal transport (OT) and develop a technique called **OTTER (Optimal TransporT adaptER)**. This OT-based approach offers a systematic way to rebalance predicted labels: data points are transported to optimal downstream classes, minimizing the overall cost in accordance with the estimated the downstream label distribution specifications.

Theoretically, we show that optimal transport given the true label distribution of the downstream can recover the Bayes-optimal classifier under mild conditions. Additionally, we provide error bounds on our adaptation method for misspecification. We provide synthetic experiments validating our theoretical claims. In real-world data settings, we validate our method on a wide variety of image and text classification tasks, showing 4.8% and 15.5% accuracy improvement on average in image and text zero-shot classification tasks, respectively. Finally, we evaluate our method in *few-shot adaptation* scenarios, where OTTER provides further improvements when combined with linear probing. Our contributions include:

- OTTER, an algorithm to deal with label distribution mismatch at inference time via optimal transport,
- Theoretical results showing the effectiveness of our method, including the ability to recover the Bayes-optimal classifier and a sensitivity analysis with respect to the label distribution specification estimation,
- Extensive empirical results on zero-shot classification for text and image datasets, showing accuracy improvements of up to 25%,
- Experimental results demonstrating the applicability of OTTER to few-shot settings, showing accuracy improvements of up to 15%, even with noisy label distribution specifications,
- Extensions of OTTER to leverage label hierarchy information or relax the batched prediction requirement and
- Application to LLM selection bias mitigation.

## 2 Background and Problem Formulation

Before presenting OTTER and providing our results, we introduce some crucial background and describe the problem setting formally.

### 2.1 Background

We briefly describe zero-shot models, the technical tool we use (optimal transport), along with other techniques that seek to address shifts. We have extended related work, including in-depth characterizations and comparisons with related methods, in Appendix B.

**Zero-shot Models.** Zero-shot classification, popularized by models such as CLIP [49], is a powerful paradigm that enables prediction on downstream tasks without additional fine-tuning. Image, language, and multimodal models have been increasingly employed for zero-shot prediction [64; 36]. These models undergo extensive pretraining on massive datasets with concept and label spaces that may be very different from those of downstream applications.

**Optimal Transport.** Optimal Transport (OT) is a framework for matching two probability distributions [48; 53]. We predominantly consider optimal transport between empirical discrete measures. Suppose that we are given points $x_1, x_2, \ldots, x_n \in \mathcal{X}$ and $y_1, \ldots, y_K \in \mathcal{Y}$, a source measure $\mu$ defined by $\mu = \sum_{i=1}^{n} w_i \delta_{x_i}$, and a target measure given by $\nu = \sum_{j=1}^{K} p_j \delta_j$, where $w_i, p_j$ are positive values such that $\sum_{i=1}^{n} w_i = 1, \sum_{j=1}^{K} p_j = 1$. Suppose also that $\delta_x$ is a Dirac delta function

---

**Algorithm 1** OTTER

---
1: **Input:** Input $\mathbf{X} = \{x_1, \ldots, x_n\}$, label distribution specification $(p_1, \ldots, p_K)$, cost matrix $C \in \mathbb{R}^{n \times K}$
2: Define input marginal $\mu = \mathbf{1}\frac{1}{n}$, prediction marginal $\nu = (p_1, \ldots, p_K)$
3: Run optimal transport and obtain transport plan $\pi$ s.t. $\pi = \arg\min_{\gamma \in \Pi(\mu, \nu)} \langle \gamma, C \rangle$.
4: Get modified classification outputs $\hat{y}_i = \arg\max_{j \in [K]} \pi_{i,j}$.
  **Return** $\{\hat{y}_i\}_{i \in [n]}$

---

at $x$, i.e. $\delta_x(x') = 1$ if $x' = x$, and $\delta_x(x') = 0$ otherwise. Given a cost matrix $C \in \mathbb{R}^{n \times K}$, the Monge-Kantorovich formulation of optimal transport is to find a minimal cost transport plan $\pi$ such that

$$\pi = \arg\min_{\gamma \in \Pi(\mu, \nu)} \langle \gamma, C \rangle,$$

where $\Pi(\mu, \nu) = \{\gamma \in \mathbb{R}_+^{n \times K} | \gamma \mathbf{1} = \mu, \gamma^T \mathbf{1} = \nu\}$.

**Distribution and Label Shifts.** Distribution shift refers to the discrepancy between a source distribution $P_s$ on which the model is trained, and a target distribution $P_t$ on which the model is deployed. Distribution shift often degrades trained model performance on target tasks. Label shift is a specific type of distribution shift such that $P_s(Y) \neq P_t(Y)$ and the data generation process is fixed — in other words, the conditional distributions of the inputs are the same: $P_s(X|Y) = P_t(X|Y)$. Techniques such as importance sampling [35; 6; 24], recalibration [3] and domain adaptation [55] are commonly used to mitigate the effects of label shift. Unfortunately, these methods assume access to source distribution data, whereas zero-shot models' pretraining data is inaccessible (often proprietary or blocked for privacy reasons). Thus, adapting zero-shot models to new label distributions poses challenges unmet by these pre-existing methods.

### 2.2 Problem Formulation

Let $\mathbf{X} = \{x_1, x_2, \ldots, x_n\}$ be an inference dataset with $x_i \in \mathcal{X}$. Furthermore, let $\mathbf{Y} = \{y_1, y_2, \ldots, y_n\}$ be the true labels of the $K$-class classification dataset, such that $y_i \in \mathcal{Y} = [K]$, are sampled according to the downstream label distribution $\nu = (p_1, p_2, \ldots, p_K)$.

Let $s_\theta(x, j) := P_\theta(Y = j | X = x)$ be a pretrained classification model constrained to the downstream label space. During pretraining, $s_\theta$ has been biased to the source label distribution $\nu^s$. We wish to offset such label distribution bias with a label distribution specification $\hat{\nu}$ for the target distribution. $\hat{\nu}$ is expected to be closer to the true label distribution of the downstream task. Given a label distribution specification, our goal is to rebalance predictions so that the predicted label distribution follows the label distribution specification.

## 3 Proposed Framework

We propose OTTER (Optimal TransporT adaptER), an optimal transport-based label distribution adaptation approach. Our goal is to have the $n$ input data points allocated to $K$ classes match a given label distribution $\hat{\nu}$, where $\sum_{j=1}^{K} \hat{\nu}_j = 1, \hat{\nu}_j \geq 0$. Specifically, we want to classify $n\hat{\nu}_1$ points as the first class, $n\hat{\nu}_2$ points as the second, and so on. However, there are many such allocations, and it is not a priori clear which one should be selected. We propose formulating an optimal transport problem that selects the allocation minimizing a particular cost:

$$\pi = \arg\min_{\gamma \in \Pi(\mu, \hat{\nu})} \langle \gamma, C \rangle,$$

where $\Pi(\mu, \hat{\nu}) = \{\gamma \in \mathbb{R}_+^{n \times K} | \gamma \mathbf{1} = \mu, \gamma^T \mathbf{1} = \hat{\nu}\}$, $\mu = \frac{1}{n}\mathbf{1}$ and $C$ is the cost (loss) matrix such that $C_{ij}$ represents a loss when we classify $x_i$ as class $j$. This procedure is described in Algorithm 1. Note that this procedure naturally matches the given label distribution specification $\hat{\nu}$.

We wish to use Algorithm 1 for zero-shot classification given the pretrained model $s_\theta$. To do so, we must select a cost function and produce $C_{ij}$. An ideal choice of such a function is $C_{ij} = -\log P_t(Y = j | X = i)$ such that optimal transport minimizes the negative log posterior under constraints. However, the target distribution $P_t$ is unknown. Instead, we replace the posterior with the classifier scores $s_\theta(x_i, j)$. We highlight that this choice of cost matrix is an natural extension of

zero-shot classification under the label distribution constraint. We prove this claim in the next section. First, we show a toy example that shows how OTTER improves zero-shot accuracy.

**Example.** To illustrate the benefits of OTTER, consider the following example for binary classification. We have two data points, $X = \{x_1, x_2\}$ with $Y = \{1, 2\}$, and true label distribution $\nu = (\frac{1}{2}, \frac{1}{2})$. Suppose that the zero-shot model's prediction scores are $s_1 = (0.4, 0.6)$ and $s_2 = (0.1, 0.9)$.

Traditional classification yields $\hat{y}_1 = 2, \hat{y}_2 = 2$, producing a 50% error rate. However, given the cost matrix $C$ derived from the prediction score matrix

$$S = \begin{bmatrix} 0.4 & 0.6 \\ 0.1 & 0.9 \end{bmatrix}, C = \begin{bmatrix} -\log 0.4 & -\log 0.6 \\ -\log 0.1 & -\log 0.9 \end{bmatrix},$$

along with $\mu = (0.5, 0.5)$ and $\nu = (0.5, 0.5)$, the optimal transport procedure discovers the transport map $\pi = \begin{bmatrix} 0.5 & 0.0 \\ 0.0 & 0.5 \end{bmatrix}$, yielding $\hat{y}_1 = 1, \hat{y}_2 = 2$. This corrects the original zero-shot prediction error.

**Extension to Online Predictions** While OTTER offers efficient label distribution adaptation, it requires a batched set of inference data points, making online predictions challenging. To address this, we introduce R-OTTER (Reweighting OTTER), which learns a reweighting factor—an estimate of $P_t(Y)/P_s(Y)$—using OTTER's predictions $y_{\text{OTTER}}$ on a validation set. Once learned, these reweighting parameters can be applied to online predictions by adjusting logit probability scores. We use a reweighting formulation equivalent to logit adjustment in [71]. The reweighted probability scores of $P_\theta$, with a reweighting vector $r \in \mathbb{R}^K$, are defined as:

$$P_{\theta,r}(Y = j|X = x) = \frac{r_j P_\theta(Y = j|X = x)}{\sum_{j'=1}^{K} r_{j'} P_\theta(Y = j'|X = x)}.$$

The parameter $r$ is learned using cross-entropy loss on $y_{\text{OTTER}}$. We expect R-OTTER to perform comparably to OTTER, with the additional advantage of not requiring OTTER to be run over the entire dataset. In the following section, we provide a theoretical result showing that $r^* = P_t(Y)/P_s(Y)$ is an optimal reweighting parameter when learned from $y_{\text{OTTER}}$ as pseudolabels, effectively addressing label shift.

## 4 Theoretical Results

In practical scenarios, label distribution specifications are frequently subject to noise, and prediction probabilities may not be well-calibrated. To understand the impact of these factors, we examine how errors in label distribution specification and calibration influence the transport plan. Our theoretical analysis yields following findings: (a) OTTER can recover the Bayes-optimal classifier in the label shift setting, (b) for a noisy cost matrix with the noisy label distribution specification setup, the suboptimality can be bounded by the deviation of cost matrix and label distribution, and (c) R-OTTER effectively addresses label shift when learned from $y_{\text{OTTER}}$ as pseudolabels.

**Classification as optimal transport.** Prior to discussing the main theoretical results, we demonstrate that standard classification—expressed as $\hat{y}_i = \arg\max_{j \in [K]} P_\theta(Y = j|X = x_i)$—can be interpreted as a (trivial) solution derived from optimal transport.

**Theorem 4.1.** *Let* $\nu_j^{ZS} = \frac{1}{n} \sum_{i=1}^{n} \mathbb{1}[\hat{y}_i^{ZS} = j]$, *where* $\hat{y}_i^{ZS} = \arg\max_{j' \in [K]} P_\theta(Y = j'|X = x_i)$. *Then, given* $C_{ij} = -\log P_\theta(Y = j|X = x_i)$,

$$\pi = \arg\min_{\gamma \in \Pi(\mu, \nu^{ZS})} \langle \gamma, C \rangle,$$

$$\hat{y}_i^{OT} = \arg\max_{j \in [K]} \pi_{ij}.$$

*Assuming there are no ties in scores, i.e.* $P_\theta(Y = j|X = x_i) \neq P_\theta(Y = j'|X = x_i)$, *for all* $j \neq j'$, *the* OTTER *predictions are equivalent zero-shot predictions, i.e.* $\hat{y}_i^{OT} = \hat{y}_i^{ZS}$ *for all* $i \in [n]$.

This theorem has the following implications. First, it suggests that the predictions will remain unchanged if we set $\hat{\nu} = \nu^{ZS}$. Second, Bayes-optimal classifiers can be derived through optimal transport, using a (true) cost matrix defined as $C_{ij}^* = -\log P_t(Y = j|X = x_i)$, coupled with the true label distribution $\nu^*$.

Our analysis begins with the label shift setup, which is a commonly-studied type of distribution shift—as well as a prominent issue when applying zero-shot models. We demonstrate that when the label distribution is correctly specified, optimal transport preserves Bayes-optimal classifier predictions under label shift. Next, we consider general perturbations in label distribution and cost matrix as well as their impact on the resulting solutions.

## 4.1 Label Shift Invariance

In this setting, we assume features follow the same conditional distribution across source and target distributions, i.e. $P_s(X|Y) = P_t(X|Y)$. Furthermore, we suppose that the prediction scores are accurately calibrated in the training dataset, such that $s_\theta(x, j) = P_s(Y = j|X = x)$. For zero-shot models, we often lack access to $P_s$. This is a typical scenario in zero-shot model applications: after training on large-scale corpora, we use the pretrained model without the source dataset.

For a given downstream task with the target label distribution $\nu^* = P_t(Y)$, one standard approach to achieve the Bayes-optimal classifier for the target distribution is to reweight the score function outputs using the ratio $\frac{P_t(Y=j)}{P_s(Y=j)}$. This adjustment leads to:

$$\tilde{s}_\theta(x, j) = s_\theta(x, j) \cdot \frac{P_t(Y = j)}{P_s(Y = j)} \propto P_t(X = x|Y = j) \cdot P_t(Y = j) \propto P_t(Y = j|X = x).$$

This reweighted score function aligns with the target distribution, thus correcting label shift.

Although reweighting the score function is a popular solution, it faces an important obstacle when applied to zero-shot models like CLIP, where the source distribution $P_s(Y)$ is typically unknown. We show that OTTER successfully induces a Bayes classifier for the target distribution, represented as $f_t(x) = \arg\max_{j \in [K]} P_t(Y = j|X = x)$, without requiring access to $P_s(Y)$. This capability is particularly significant for zero-shot models, enabling them to adapt to target distributions effectively, even in the absence of explicit knowledge of the source distribution.

Now, we show that optimal transport can be an effective tool to correct label shift.

**Theorem 4.2.** *Suppose the pretrained model is well-calibrated for the source distribution,*

$$P_\theta(Y = j|X = x_i) = P_s(Y = j|X = x_i)$$

*and there is no tie probability, for all $j \neq j', i \in [n]$*

$$P_\theta(Y = j|X = x_i) \neq P_\theta(Y = j'|X = x_i).$$

*Denote the Bayes optimal predictions in the target distribution as $\hat{y}_i^* = \arg\max_{j \in [K]} \log P_t(Y = j|X = x_i)$. Then* OTTER *predictions $\hat{y} = $* OTTER$(\mathbf{X}, \nu^*, C)$ *are the same as Bayes optimal predictions $\hat{y}^*$.*

That is, OTTER recovers a Bayes classifier in the target distribution without access to the source distribution, given the target distribution and a well-calibrated model for the source dataset.

## 4.2 General Perturbation Sensitivity

In practical applications, calibration error could extend beyond noise in the elements of the cost matrix. A key source of error is label distribution estimation error. Hence, we address a more general setting, examining the impact of simultaneous perturbations in the label distribution and cost matrix of the transport plan. Our result applies techniques from perturbation theory for linear programming .

We rewrite our optimal transport problem $\min_{\pi \in \Pi(\mu,\nu)} \langle \pi, C \rangle$ as a linear programming problem. Let $\pi$ and $C$ be the transport plan and cost matrix respectively. Matrix $G$ and vector $g$ are used to denote the row and column constraints on $\pi$ to form a feasible plan which transports distribution from $\mu$ to $\nu$.

$$H := \begin{bmatrix} \mathbf{1}_n^T \otimes \mathbb{I}_K \\ \mathbb{I}_n \otimes \mathbf{1}_K^T \end{bmatrix}, G = \begin{bmatrix} H \\ -H \end{bmatrix}, g = \begin{bmatrix} \mu \\ \nu \\ -\mu \\ -\nu \end{bmatrix}.$$

Then, we have the equivalent linear programming problem,

$$\min \left\{ \sum_{i,j} C_{i,j} \pi_{i,j} | G \cdot \text{vec}(\pi) \geq g, \pi \geq 0 \right\}. \tag{1}$$

We adapt a theorem from Robinson [52] with our optimal transport problem notation.

**Theorem 4.3.** *Let the primal linear programming problem be defined as in equation (1), and its dual problem be $\max\{w^T g | w^T G \leq vec(C)^T, w \geq 0\}$. Suppose perturbed cost matrix is $\hat{C} = C + \Delta_C$, the perturbed class distribution $\hat{\nu} = \nu + \Delta_\nu$, such that $\hat{g} = g + \Delta_g$ where*

$$\Delta_g = \begin{bmatrix} 0 \\ \hat{\nu} - \nu \\ 0 \\ -\hat{\nu} + \nu \end{bmatrix}.$$

*Assume that primal and dual problems are solvable. Denote the original solutions as $\pi, w$ and perturbed solutions as $\hat{\pi}$ and $\hat{w}$. Then,*

$$\|\pi - \hat{\pi}\|_F^2 \leq \kappa^2 \left( \|\Delta_\nu\|_2^2 + \|[vec(\Delta_C)]_+\|_2^2 + \|vec(C)^T vec(\hat{\pi}) - g^T \hat{w}\|_2^2 \right) - \|w - \hat{w}\|_2^2$$

*, where $1 \leq p \leq \infty$ and $\kappa$ is a Hoffman constant that only relates to the original problem [30].*

Ignoring the constant and the subtraction part, the upper bound can be decomposed into three components,

- $\Delta_\nu$: noise (or the estimation error) of the target balance,

- $[vec(\Delta_C)]_+$: noise (or the calibration error) of the cost matrix,

- $vec(C)^T vec(\hat{\pi}) - g^T \hat{w}$: the suboptimality of perturbed solution $\hat{w}$.

Theorem 4.3 implies that the deviation from perturbed solution to true solution is bounded by the magnitude of perturbations and suboptimality of the perturbed solution. From this result, we can expect prediction accuracy to deteriorate with perturbations in the label distribution and calibration.

### 4.3 Effectiveness of R-OTTER

We provide a theoretical result showing that R-OTTER can learn an optimal parameter by learning reweighting parameters from $\hat{y}_{\text{OTTER}}$ as pseudolabels, and produce identical predictions to OTTER once the optimal parameter is obtained in the label shift setup.

**Theorem 4.4.** *Under the same assumptions as in Theorem 4.2, the parameter $r^* = P_t(Y)/P_s(Y)$ is optimal when learning with $y_{\text{OTTER}}$ as pseudolabels.*

The proof is provided in Appendix D.6.

## 5 Experiments

The primary objective of our experiments is to (1) validate that OTTER improves zero-shot model performance when given accurate label distribution estimates and (2) investigate its sensitivity to perturbations. In experiments on real datasets (Section 5.1), we confirm that OTTER can improve zero-shot classification significantly in a variety of settings. In synthetic experiments (Section 5.2), we validate our theoretical claims—label shift invariance and sensitivity to perturbation in a fully controllable setting. Additionally, we show that OTTER can be combined with label distribution estimation methods in the few-shot learning setting (Section 5.3). Next, we show the empirical results for H-OTTER that leverages label hierarchy (Section 5.4) and R-OTTER that mitigates the batched prediction requirement (Section 5.5). Finally, we we apply OTTER to mitigate LLM selection bias (Section 5.6). Our code is available at https://github.com/SprocketLab/OTTER.

### 5.1 Real Data Experiments

We hypothesize that the model performance can improve significantly when the label distribution specification is exact.

**Setup and Procedure.** We used 17 image classification datasets and 4 text classification datasets. We employed CLIP [49] for image zero-shot classification, and BERT [19]. A comprehensive list and details of experiments can be found in Appendix E.

**Baseline.** We adopt Prior Matching (PM) [37] as a baseline. It optimizes score weighting parameters to align with the label distribution specification. A detailed explanation of Prior Matching is given in Appendix C. It is worth noting that the performance of *Prior Matching is highly sensitive to hyperparameters such as temperature and learning rate*. Optimal hyperparameters may vary across

| | Zero-shot | Prior Matching | OTTER | | Zero-shot | Prior Matching | OTTER |
|---|---|---|---|---|---|---|---|
| CIFAR10 | 88.3 | 91.3 ($\pm$0.0) | **91.7** | Oxford-IIIT-Pet | 83.8 | 82.0 ($\pm$0.3) | **88.8** |
| CIFAR100 | 63.8 | 64.1 ($\pm$2.7) | **67.9** | Stanford-Cars | 55.7 | 39.8 ($\pm$2.6) | **59.7** |
| Caltech101 | 79.8 | 59.3 ($\pm$15.4) | **88.7** | STL10 | 98.0 | 98.4 ($\pm$0.0) | **98.6** |
| Caltech256 | 79.8 | 9.5 ($\pm$1.5) | **87.0** | SUN397 | 47.1 | 6.7 ($\pm$1.6) | **54.1** |
| Country211 | 19.8 | 19.0 ($\pm$0.1) | **21.1** | CUB | 46.0 | 40.4 ($\pm$0.0) | **50.4** |
| DTD | 39.0 | 42.1 ($\pm$0.1) | **44.4** | ImageNet | 60.2 | 53.6 ($\pm$0.1) | **62.9** |
| EUROSAT | 32.9 | 41.6 ($\pm$0.8) | **42.0** | ImageNet-r | 68.9 | 16.7 ($\pm$3.5) | **72.4** |
| Flowers102 | 64.0 | 54.0 ($\pm$14.1) | **70.8** | ImageNet-Sketch | 39.8 | 36.5 ($\pm$0.4) | **44.5** |
| Food101 | 85.6 | 86.8 ($\pm$3.1) | **89.9** | | | | |
| Amazon review | 74.0 | 58.8 ($\pm$46.4) | **91.7** | GenderBias | 84.1 | 41.4 ($\pm$39.6) | **91.9** |
| CivilComments | 48.4 | 57.2 ($\pm$37.7) | **81.4** | HateXplain | 30.9 | 31.3 ($\pm$3.3) | **34.3** |

Table 1: Accuracy (%) in zero-shot image classification (ViT-B/16) and text classification (BERT). We use the true label distribution as the label distribution specification. The numbers in parenthesis of Prior Matching represent the standard deviation of 10 different samplings of the validation set. OTTER produces improvements nearly across-the-board, with an average lift 4.9% in image classification and 15.5% in text classification, outperforming a powerful baseline, prior matching in almost all cases.

different datasets. We selected hyperparameters through grid search, by evaluating their performance on a validation set, consisting of 10 labeled examples per class. In contrast, we highlight that OTTER is tuning-free.

**Results.** Table 1 shows the image classification results with CLIP (ViT-B/16) and the text classification results with BERT. Notably, OTTER *demonstrates a 4.8% and 15.5% enhancement on average in image and text zero-shot classification, respectively*. While Prior Matching shows competitive performance when accurately tuned, it often struggles. We found that hyperparameter tuning fails in the class-imbalanced datasets such as Caltech256, SUN397, ImageNet-r (Appendix E, Table 7). This suggests that the hyperparameter selection process necessitates a validation set label distribution similar to the target distribution—rendering it unusable in zero-shot scenarios. More details and additional experiment results — including the sensitivity study on the label distribution specification error, computation time, and combination with other prompting methods — are provided in Appendix E.3.

## 5.2 Synthetic Experiments

We hypothesize OTTER is invariant to label shift under the conditions in Theorem 4.2. We also investigate the sensitivity to perturbations of the cost matrix and the label distribution.

**Setup and Procedure.** We simulate label shift in logistic regression. Suppose $X|Y = 0 \sim \mathcal{N}(-1, 1)$ and $X|Y = 1 \sim \mathcal{N}(1, 1)$. Training data is sampled from a mixture of Gaussians $X_s \sim \nu_0^s \mathcal{N}(-1, 1) + \nu_1^s \mathcal{N}(1, 1)$ such that $P_s(Y = 0) = \nu_0^s, P_s(Y = 1) = \nu_1^s, \nu_0^s + \nu_1^s = 1$. Similarly, we sample the test data from $X_t \sim \nu_0^t \mathcal{N}(-1, 1) + \nu_1^t \mathcal{N}(1, 1)$. We fix the training set label distribution as $\nu_0^s = 0.1, \nu_1^s = 0.9$ and vary test set label distribution $\nu^t$ to simulate label shift. We train a logistic regression model with 10,000 samples from the source distribution, and test the model with 10,000 samples from the target distribution. A Bayes-optimal classifier in the target distribution is given by $f_{Bayes}(x) = \mathbb{1}[x \geq \frac{1}{2}(\log \frac{\nu_0^t}{\nu_1^t} + 1)]$. The naive classifier is defined as the maximizer of the predicted score. The OTTER predictions are produced with Algorithm 1, with the cost matrix $C_{ij} = -\log P_\theta(Y = j|X = x_i)$ and the label distribution specification $\nu^t$, where $P_\theta(Y|X)$ represents the logistic regression model scores.

We separately investigate perturbed prediction score matrix and perturbed label distribution specification's impact on the prediction accuracy. For perturbed prediction scores, we fix the label distribution to be the true one, and add noise $\delta \sim \mathcal{N}(0, \sigma^2)$ of varying levels $\sigma$ to the predicted score $P_\theta(Y = 1|X)$. For the perturbed label distribution specification, we fix the prediction score to be true scores and add noise $\epsilon$: $\hat{\nu} = \nu^t + (\epsilon, -\epsilon)$. We use these perturbed variants to obtain perturbed solutions and compare with ground-truth solution.

**Results.** Figure 2 illustrates how accuracy changes with label shift when the predicted score is perturbed and when label distribution specification is perturbed. We observe that the naive classifier

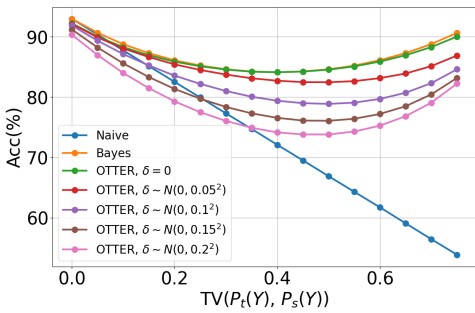 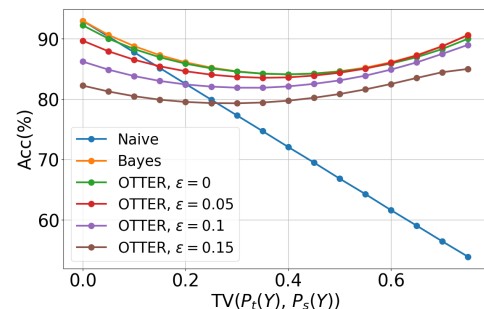

(a) Prediction accuracy changes with perturbed confidence score.

(b) Prediction accuracy changes with perturbed label distribution.

Figure 2: Synthetic experiment results. X-axis represents total variation distance between the source and the target distribution, describing label shift severity. Y-axis represents prediction accuracy. Curves represent different methods and noise levels. Our approaches dramatically outperform the baseline at higher mismatch levels.

deteriorates as the total variation distance between source and target distributions increases. It indicates that naive classifier is sensitive to label shift. However, without perturbation, OTTER *remains unaffected by the label distribution shift*, which validates our invariance result in Section 4.

In the case of confidence prediction perturbation, both the naive classifier and OTTER have accuracy decreasing as perturbation level increases. For simplicity, we omitted the naive classifier's performances under different levels of noise as adding zero-mean noise does not alter its accuracy significantly. We observe that OTTER has better performance than the naive method when significant label shift exists. Similarly, for label distribution perturbation, we observe as the noise level $\epsilon$ increases, OTTER's accuracy downgrades—but still yields better performance when label shift is severe.

Our experimental results suggest simply using prediction scores for zero-shot classification leads to inaccurate predictions under label shift, while OTTER is robust to label shift when no perturbations are present. Perturbations in both predicted score and label distribution specification downgrades the predicted accuracy, as expected, but OTTER still yields better results than the naive baseline.

## 5.3 Few-shot adaptation with label distribution estimation

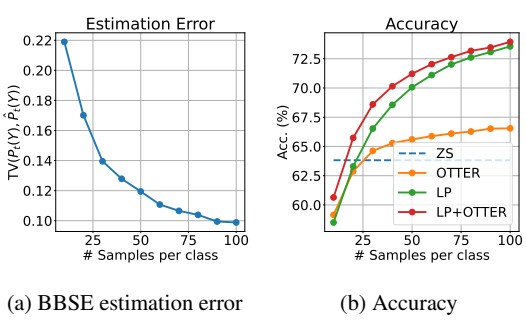

(a) BBSE estimation error     (b) Accuracy

Figure 4: Ablation on the number of samples in few-shot learning. In (a), We can observe that BBSE estimation get more precise as the number of samples increases. Following this, OTTER gets better accuracy in (b). Additionally, OTTER consistently improves linear probing when combined.

We anticipate that OTTER can be used in few-shot learning when combined with label distribution estimation methods. We expect OTTER can improve zero-shot classification if the label distribution estimation error is sufficiently small. Also, we expect OTTER can improve linear probing, which is one of standard approaches for few-shot learning.

**Setup and Procedure.** We use the same datasets as the previous experiment. We consider a 10-shot learning setting: 10 labeled samples per class are given. Note that labeled samples have uniform label distribution, while the label distribution in the target distribution *may not be uniform*. This setting requires the use of label distribution estimation methods used in label shift adaptation [35; 6; 24]. We estimate the target label

| Dataset | ZS | ZS BBSE+PM | ZS BBSE+OT | LP | LP BBSE+PM | LP BBSE+OT |
|---|---|---|---|---|---|---|
| CIFAR10 | 88.3 | 72.7 | 87.5 | **90.2** | 89.8 | 90.0 |
| CIFAR100 | **63.8** | 3.2 | 59.1 | 58.3 | 24.4 | 60.5 |
| Caltech101 | 79.8 | 32.5 | 80.7 | **91.5** | 87.5 | 91.4 |
| Caltech256 | 79.8 | 6.0 | 80.3 | 84.5 | 58.4 | **85.4** |
| Country211 | **19.8** | 1.5 | 15.9 | 12.4 | 9.2 | 13.2 |
| DTD | 39.0 | 3.2 | 31.2 | 58.6 | 49.0 | **59.3** |
| EUROSAT | 32.9 | 19.2 | 34.0 | 74.6 | 71.6 | **75.9** |
| Flowers102 | 64.0 | 40.3 | 60.8 | 89.0 | 87.8 | **90.2** |
| Food101 | **85.6** | 15.3 | 82.3 | 79.1 | 60.6 | 79.8 |
| Oxford-IIIT-Pet | **83.8** | 43.3 | 71.4 | 75.7 | 72.0 | 75.6 |
| Stanford-Cars | 55.7 | 2.3 | 51.7 | 64.5 | 65.4 | **66.3** |
| STL10 | **98.0** | 97.4 | 96.9 | 97.7 | 97.5 | 97.6 |
| SUN397 | **47.1** | 6.9 | 25.6 | 0.2 | 0.2 | 0.2 |
| cub | 46.0 | 3.3 | 45.5 | 72.2 | 63.3 | **75.6** |
| ImageNet | **60.2** | 0.8 | 57.7 | 56.8 | 53.6 | 59.8 |
| ImageNet-r | **68.9** | 1.7 | 63.3 | 54.9 | 47.6 | 57.1 |
| ImageNet-Sketch | 39.8 | 0.8 | 40.4 | 43.4 | 37.9 | **48.3** |
| Amazon | 74.0 | 47.9 | **89.1** | 71.3 | 66.9 | 71.3 |
| CivilComments | 48.3 | **69.1** | 55.8 | 53.8 | 45.5 | 53.8 |
| Gender | 84.0 | 57.0 | **87.8** | 78.0 | 71.2 | 78.5 |
| HateXplain | 30.4 | 34.4 | **35.2** | 32.8 | 32.7 | 32.3 |

Table 2: Accuracy (%) with OTTER combined with class balance estimation. ZS BBSE denotes BBSE label distribution estimation based on zero-shot prediction scores, and LP BBSE denotes BBSE label distribution estimation based on linear probing prediction scores. We report the mean of 10 different random samplings of the validation set. OTTER produces moderate improvements when comined with linear probing in image classification tasks. In text classification tasks, OTTER significantly improves accuracy, up to 15.1%, even with noisy label distribution estimation.

distribution with Black Box Shift Estimation (BBSE) [35]. BBSE estimates the target balance using confusion matrix, under the label shift assumption. For detailed explanation, refer to Appendix C.

**Results.** Table 2 shows the image and text zero-shot classification results with the label distribution estimation via BBSE and linear probing. The image classification results show that OTTER can yield mild improvement over linear probing, even with the label distribution estimation errors. Figure 4 shows that accuracy improvement is consistent across the number of samples used for linear probing. In text classification, we found OTTER improves zero-shot text classifications where the number of classes is small ($K = 2$ or $3$). While it shows a relatively high variance due to the small sample size ($20 \sim 30$), the average accuracy improves significantly over zero-shot classification. More detailed analysis regarding label distribution estimation error and the number of samples is provided in Appendix E.4.

## 5.4 Zero-shot prediction improvement with class hierarchy

| | OTTER | H-OTTER |
|---|---|---|
| RN50 | 38.5 ($\pm$4.9) | **43.6** ($\pm$3.1) |
| RN101 | 39.9 ($\pm$6.9) | **44.8** ($\pm$5.1) |
| ViT-B/32 | 59.0 ($\pm$3.1) | **59.3** ($\pm$2.9) |
| ViT-B/16 | 54.6 ($\pm$8.3) | **58.2** ($\pm$3.6) |
| ViT-L/14 | **71.3** ($\pm$3.9) | 69.4 ($\pm$5.2) |

Table 3: Accuracy (%) with hierarchical OTTER (H-OTTER). (H-OTTER) yields additional improvements over OTTER, up to 5.1%, using the hierarchy information of labels.

We hypothesize incorporating class hierarchy information can enhance few-shot label distribution estimation and thus improve zero-shot predictions.

**Setup and Procedure.** We use a subset of CIFAR100 data with WordNet hierarchy. Specifically, we take 'fish' and 'tree' as superclasses and have 5 subclasses in each of them. We suppose we can access 10 labeled samples per each subclass. We first apply OTTER with the superlevel label distribution estimation and make pseudo-labels of superlevel class in the test set. Using them, we estimate the sublevel label distribution and use OTTER.

**Results.** Table 3 presents the results. As anticipated, we note an enhancement in accuracy when compared to the naive implementation of OTTER. Specifically, we observe a significant improvement in accuracy for RN50, RN101, and ViT-B/16,

| | Zero-shot | OTTER | R-OTTER | | Zero-shot | OTTER | R-OTTER |
|---|---|---|---|---|---|---|---|
| CIFAR10 | 88.3 | 91.7 | 88.4 | Oxford-IIIT-Pet | 83.8 | 88.8 | 85.7 |
| CIFAR100 | 63.8 | 67.9 | 65.3 | Stanford-Cars | 55.7 | 59.7 | 51.0 |
| Caltech101 | 79.8 | 88.7 | 88.2 | STL10 | 98.0 | 98.6 | 98.1 |
| Caltech256 | 79.8 | 87.0 | 79.6 | SUN397 | 47.1 | 54.1 | 46.6 |
| Country211 | 19.8 | 21.1 | 19.5 | CUB | 46.0 | 50.4 | 44.4 |
| DTD | 39.0 | 44.4 | 44.0 | ImageNet | 60.2 | 62.9 | 59.8 |
| EUROSAT | 32.9 | 42.0 | 39.3 | ImageNet-r | 68.8 | 72.4 | 68.5 |
| Flowers102 | 64.0 | 70.8 | 69.7 | ImageNet-Sketch | 39.8 | 44.5 | 39.3 |
| Food101 | 85.6 | 89.9 | 88.5 | | | | |

Table 4: Accuracy (%) of naive zero-shot, OTTER, and R-OTTER in zero-shot image classification (ViT-B/16)

| | ARC-Challenge | | | | CommonsenseQA (CSQA) | | | | MMLU | | | |
| | Naive | | OTTER | | Naive | | OTTER | | Naive | | OTTER | |
| Model | Acc.($\uparrow$) | RStd ($\downarrow$) | Acc. | RStd | Acc. | RStd | Acc. | RStd | Acc. | RStd | Acc. | RStd |
|---|---|---|---|---|---|---|---|---|---|---|---|---|
| Llama-2-7b | 36.0 | 27.4 | **45.5** | **1.7** | 31.9 | 28.4 | **42.7** | **3.8** | 36.1 | 22.5 | **40.0** | **0.9** |
| Llama-2-13b | **62.9** | 6.0 | 62.8 | **1.5** | 57.0 | 10.2 | **58.1** | **2.0** | 51.1 | 6.9 | **51.8** | **1.3** |
| Llama-2-7b-chat | 56.5 | 12.4 | **57.4** | **1.3** | 56.5 | 15.2 | **60.4** | **3.5** | 45.9 | 13.1 | **46.6** | **0.5** |
| Llama-2-13b-chat | 64.4 | 13.7 | **66.2** | **2.5** | 64.0 | 9.8 | **66.4** | **1.8** | 52.3 | 13.9 | **53.8** | **1.0** |
| vicuna-7b | 53.5 | 8.6 | **54.1** | **2.3** | 56.9 | 8.3 | **57.6** | **1.0** | 46.6 | 5.8 | **46.9** | **0.3** |
| vicuna-13b | 62.9 | 8.3 | **63.3** | **2.4** | 63.4 | 12.9 | **64.5** | **3.1** | 50.5 | 9.9 | **50.7** | **1.1** |

Table 5: Mitigation of selection bias via OTTER.

which we attribute primarily to the reduction in label distribution estimation error. Further details are provided in Appendix E.5.

### 5.5 Effectiveness of R-OTTER

We show that R-OTTER provides a performance comparable to that of OTTER empirically.

**Setup and Procedure.** We use the identical setup for Section 5.1. R-OTTER learns reweighting parameters in validation set using $y_{\text{OTTER}}$. Note that the validation set is not required to be labeled since $y_{\text{OTTER}}$ is used as pseudolabels in the validation set.

**Results.** Although R-OTTER is suboptimal compared to OTTER due to generalization issues, it still provides label distribution correction, improving accuracy over zero-shot predictions. We also provide synthetic experiments for R-OTTER in Appendix E.6.

### 5.6 Mitigating selection bias in LLM multiple-choice questions

Selection bias is the tendency of LLMs to favor certain prefix tokens in multiple-choice questions [69; 17; 12; 65]. We demonstrate that OTTER can effectively mitigate selection bias by randomly shuffling the options and enforcing a uniform class balance in OTTER.

**Setup and Procedure.** Our experimental setup follows Zheng et al. [69], utilizing the MMLU [28], ARC-Challenge [15], and CommonsenseQA (CSQA) [57] datasets. We test with llama-2(-chat)-7/13B [60] and vicuna-v1.3-7/13B [11] models, treating the probabilities of each option token (A/B/C/D) as prediction probabilities. OTTER is applied under the assumption of a uniform distribution. We use 0-shot predictions and evaluate performance using accuracy and the standard deviation of recalls (RStd) as metrics.

**Results.** Table 5 presents the experimental results. OTTER significantly reduces selection bias, enhancing accuracy by up to 10.6% and lowering RStd by as much as 25.7%.

## 6 Conclusion

While zero-shot models have been successful, pretraining using Internet-scale datasets yields artifacts that may harm downstream tasks. In this paper, we identify the bias in class balance, and provide a simple but powerful solution using optimal transport. Theoretically, we describe how OT can fix label distribution mismatch and its sensitivity to perturbations. Empirically, we validated our approach's ability to improve zero-shot classification accuracy, mitigating label distribution mismatch in zero-shot models. We believe our method can expedite the deployment of zero-shot classification, reducing the necessity of finetuning for downstream tasks.

**Acknowledgments**

We are grateful for the support of the NSF under CCF2106707 (Program Synthesis for Weak Supervision) and the Wisconsin Alumni Research Foundation (WARF). Jitian Zhao gratefully acknowledge support from the IFDS at UW-Madison and NSF through TRIPODS grant 2023239 for their support. We thank Nick Roberts for his valuable discussions and insights.

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

# Appendix

The appendix contains glossary, algorithm details, proofs, and detailed experimental results. The glossary contains a convenient reminder of our terminology (Appendix A). Appendix B provides more related works and discussion about the relationship between our work and related papers. Appendix C describes the relevant algorithms used in our experiments, including Prior Matching [37] and BBSE [35]. Appendix D provides the proofs of theorems that appeared in Section 4. Finally, we give more details and analysis of the experiments and provide additional experiment results in Appendix E.

## A Glossary

The glossary is given in Table 6 below.

| Symbol | Definition |
|---|---|
| $n$ | Number of points |
| $K$ | Number of classes |
| $[K]$ | The set of classes $\{1, 2, \ldots, K\}$ |
| $\mathcal{X}$ | Input feature space |
| $\mathcal{Y}$ | Label space |
| $X$ | Input features |
| $Y$ | True labels |
| $P_s$ | Source (training) distribution of data |
| $P_t$ | Target (testing) distribution of data |
| $s_\theta$ | Prediction score function with parameter $\theta$ |
| $C^*$ | Bayes optimal cost matrix for prediction |
| $\hat{C}$ | Estimate of cost matrix for prediction |
| $\nu$ | Class balance for true labels |
| $\nu^{ZS}$ | Class balance for predicted labels from the zeroshot model |
| $\Delta_C$ | Additive perturbations to cost matrix |
| $\Delta_\nu$ | Additive perturbation to class balance |
| $\pi$ | Optimal transport plan |
| $G, g$ | Constraint matrix and vector for linear programming s.t. feasible set is $\{x \in \mathcal{X} : Gx \geq g\}$ |
| $w$ | Dual solution for linear programming problem |
| $\kappa$ | Hoffman constant for the true optimal transport problem |
| $[x]_+$ | Positive parts of $x$, i.e. $[x]_+ := x\mathbb{1}[x > 0]$ |
| $[x]_-$ | Negative parts of $x$, i.e. $[x]_- := x\mathbb{1}[x < 0]$ |
| $\text{vec}(A)$ | Vectorized $A$, $\text{vec}(A) = [A_{11}, \ldots A_{m1}, A_{12}, \ldots, A_{m2}, \ldots, A_{1n}, \ldots A_{mn}]^T$ for $A \in \mathbb{R}^{m \times n}$ |

Table 6: Glossary

## B Extended Related Work

**Improving Zero-shot Classification at Inference Time.** As zero-shot classification has gained popularity, several works have been developed to improve zero-shot classification at inference time. Chuang et al. [13]; Adila et al. [1] use vector projection methods to remove spurious correlations at inference time. Menon and Vondrick [40]; Novack et al. [43]; An et al. [4] augment prompts with language models and combine their classification output to improve zero-shot performance. Roberts et al. [51] uses additional information of label space geometry to extend model pre-trained on the label subset to broader use-cases. While these works try to improve zero-shot classification at inference time in common, the main difference is that our method tackles the inherent class prior of zero-shot models.

**Label Shift Adaptation.** Label shift adaptation methods are designed to address the negative impacts arising from changes in the label distribution. These methods typically follow a two-step process [35; 6; 24]. The first step involves estimating the label distribution within the target dataset using labeled data from the source distribution and unlabeled data from the target distribution. Next, the prediction scores are reweighted using the estimated target label distribution and the source label

distribution. However, the standard approach requires access to the labeled source distribution data, which is usually not possible in zero-shot classification scenarios. OTTER provides a solution decoupled from the source data distribution, overcoming this limitation.

**Improving Zero-shot Classification using Prior.** Several studies have explored leveraging prior information to enhance zero-shot classification, even in the absence of access to source distributions. In the context of prompt-based zero-shot models, prior matching [37] employs word prior distribution to alleviate word bias inherent in pretraining data. We adopted their adaptation method as a baseline. Similarly, Kahana et al. [31] develop adaptation layers trained using priors, aiming to maintain proximity to the original scores. However, both approaches entail training additional layers and necessitate hyperparameter tuning, which may pose challenges in the context of zero-shot predictions. In contrast, OTTER presents a straightforward and efficient adaptation method to new label distributions without the need for any hyperparameter tuning, backed by theoretical guarantees.

**Leveraging Optimal Transport for Enhanced Pseudo-labeling and Classification.** A number of studies have explored the enhancement of pseudo-labeling and classification tasks through optimal transport, using label distribution specifications, in a similar spirit to our work, but within different contexts. Tai et al. [56] uses optimal transport to allocate pseudo-labels to unlabeled datasets based on the label distribution specification in the semi-supervised setup. Wang et al. [62] deals with long-tail distribution in the partial-label learning setup based on optimal transport. Zhang et al. [68] uses partial optimal transport as a pseudo-labeling based approach for deep imbalanced clustering, progressively expanding the labeled sample proportion. Guo et al. [26] reweights the training dataset to match the label distribution specification using optimal transport. This work mainly deals with the class imbalance problem in the training step. Shi et al. [54] studies classification from a matching perspective, revealing the connection between softmax loss and inverse optimal transport and suggesting a new loss function to address long-tail distributions. Their analysis provides useful insights for OTTER — why cost matrix induced by pre-trained models can be useful in the inference step. Xian et al. [66] uses optimal transport as a postprocessing method to guarantee demographic parity. While sharing aspects of the approach, our work addresses class bias in zero-shot models. Peng et al. [47] used optimal transport to handle long-tail recognition with a learned cost matrix. Our study provides a theoretical basis for understanding their empirical results. Chang et al. [9] employs optimal transport to detect common and private classes between the source and the target domain, under the universal domain adaptation setting, where knowledge is transferred from source to target domain without any constraints on the label sets. In the context of zero-shot classification, there is no need to manage label space disparities between the source and target domains. Instead, the main concern of zero-shot classification is dealing with the distribution shift between the pretraining dataset and the downstream task. We tackle the label distribution shifts using optimal transport.

**Class Imbalance.** Class imbalance problems occur when the number of instances across different classes in a dataset is disproportionately distributed. This imbalance can severely bias the traininig process of a machine learning model, leading to poor generalization performance, especially for the minority classes. It has been extensively studied in the context of traditional machine learning [23]. Oversampling [10] and cost-sensitive learning [58] are well-known approaches to address class imbalance. Nonetheless, the inherent nature of class imbalance in pretraining datasets introduces a distinct set of challenges, especially when attempting to rectify such biases within the context of zero-shot classification scenarios.

## C Algorithm details

**Prior matching** [37] proposed prior matching as a reweighting method for prompt-based classifiers to mitigate word bias — the distribution shift between pretrained language models' word prior and the class priors in the downstream task. We use it as a zero-shot model adaptation method given a class balance estimation.

Define reweighted probability scores of $P_\theta$ with $r$ as $P_{\theta,r}(Y = j|X = x_i) = \dfrac{r_j P_\theta(Y = j|X = x_i)}{\sum_{j'=1}^{K} r_{j'} P_\theta(Y = j'|X = x_i)}$. Ideally, we hope to estimate the weight vector $r^* \in \mathbb{R}^n$ such that reweighted scores $P_{\theta,r^*}(Y = j|X = x_i)$ maximizes the accuracy in the target distribution. Since the labels are not given, this is impossible. Instead, prior matching matches the label distribution of

**Algorithm 2** Black Box Shift Estimator (BBSE)

---

**Input:** Source input data $\mathbf{X^s} = \{x_1^s, \ldots, x_m^s\}$, Source labels $\mathbf{Y^s} = \{y_1^s, \ldots, y_m^s\}$, Target input data $\mathbf{X^t} = \{x_1^t, \ldots, x_n^t\}$, model prediction distribution $P_\theta$

Estimate the source class balance $\nu^s$ such that $\nu_j^s = \dfrac{\sum_{i=1}^n P_\theta(Y = j | X = x_i^s)}{m}$

Compute the naive target class balance $\tilde{\nu}^t$ such that $\tilde{\nu}_j^t = \dfrac{\sum_{i=1}^n P_\theta(Y = j | X = x_i^t)}{n}$

Estimate confusion matrix $V$ such that $A_{jk} = \dfrac{1}{m} \sum_{i=1}^m P_\theta(Y = k | X = x_i^s)$

Estimate the refined target class balance $\hat{\nu}^t = A \tilde{\nu}^t$

**Return** $\hat{\nu}$

---

predicted classes with the class balance estimate $\hat{\nu}$, i.e.

$$\hat{r}_j = \arg\min_{r_j \in \mathbb{R}} \left| \sum_{i=1}^n P_{\theta,r}(Y = j | X = x_i) - \nu_j \right|.$$

It can be solved using the standard optimization techniques — we used [38]. While this is equivalently effective with OTTER when properly optimized, we found that the temperature parameter and learning rate crucially affect the final accuracy, making it less ideal for the zeroshot adaptation. We used the grid search with the small validation set (10 samples per class) in each task to select hyperparameters. The hyperparameter ranges are as follows.

- Temperature: [1e-3, 1e-4, 1e-5, 1e-6, 1e-7]

- Learning rate: [1e-3, 1e-4, 1e-5, 1e-6, 1e-7]

**Black Box Shift Estimation (BBSE)**    Label shift adaptation methods [35; 6; 24] aims to estimate the class balance in the target distribution using the labeled source distribution data and the unlabeled target distribution data. We use Black Box Shift Estimation (BBSE) to estimate the class balance in the downstream task. Algorithm 2 describes the procedure. Note that the derivation of this algorithm depends on the label shift assumptions, thus the label distribution estimation with real-world data can be heavily biased.

# D   Theory details

## D.1   Proof of Theorem 4.1

*Proof.* Suppose $\hat{y}_i^{OT} \neq \hat{y}_i^{ZS}$ for some $i \in [n]$. It implies $\sum_{i=1}^n -\log P_\theta(Y = \hat{y}_i^{OT} | X = x_i) < \sum_{i=1}^n -\log P_\theta(Y = \hat{y}_i^{ZS} | X = x_i)$. However, this is a contradiction since, for any $i \in [n]$, $\hat{y}_i^{ZS} = \arg\max_{j \in [K]} P_\theta(Y = j | X = x_i)$, thus $-\log P_\theta(Y = \hat{y}_i^{ZS} | X = x_i) \leq -\log P_\theta(Y = j | X = x_i)$ for all $j \in [K]$, which results in $\hat{y}_i^{OT} = \hat{y}_i^{ZS}$. $\qquad\square$

## D.2   Proof of Theorem 4.2

To prove Theorem 4.2, we show a specific form of invariance property of optimal transport first.

**Theorem D.1.** *Suppose $\pi^* = \arg\min_{\gamma \in \Pi(\mu, \nu)} \langle \gamma, C \rangle$ and $E$ is a columnwise perturbation, i.e.,*

$$E = \begin{bmatrix} \epsilon_1 \mathbf{1} & \epsilon_2 \mathbf{1} & \cdots \epsilon_K \mathbf{1} \end{bmatrix},$$

*where $\mathbf{1}$ denotes n dimensional vectors and $\epsilon_1, \ldots, \epsilon_K$ are constants. Then the perturbed cost matrix $\tilde{C} = C + E$, then $\pi^*$ is also an optimal transport map with respect to the cost matrix $\tilde{C}$.*

*Proof.* By the optimality condition, we have

$$\sum_{i,j} \pi_{ij}^* C_{ij} \leq \sum_{i,j} \pi_{ij} C_{ij}$$

for any $\pi \in \Pi(\mu, \nu)$. Then,

$$\sum_{i,j} \pi_{ij}^* C_{ij} + \sum_{j=1}^K \nu_j \epsilon_j \leq \sum_{i,j} \pi_{ij} C_{ij} + \sum_{j=1}^K \nu_j \epsilon_j,$$

which is

$$\sum_{i,j} \pi_{ij}^* C_{ij} + \sum_{j=1}^{K} \sum_{i=1}^{n} \pi_{ij}^* \epsilon_j \leq \sum_{i,j} \pi_{ij} C_{ij} + \sum_{j=1}^{K} \sum_{i=1}^{n} \pi_{ij} \epsilon_j.$$

Thus,

$$\sum_{i,j} \pi_{ij}^* \tilde{C}_{ij} \leq \sum_{i,j} \pi_{ij} \tilde{C}_{ij}.$$

$\square$

This theorem is also valid for row-wise perturbations as well with a similar proof. Consequently, a straightforward implication is that

**Corollary D.2.** *Suppose $\pi^* = \arg\min_{\gamma \in \Pi(\mu,\nu)} \langle \gamma, C \rangle$, $E$ is a columnwise perturbation and $F$ is a row-wise perturbation, such that*

$$E = \begin{bmatrix} \epsilon_1 \mathbf{1} & \epsilon_2 \mathbf{1} & \cdots \epsilon_K \mathbf{1} \end{bmatrix},$$

$$F = \begin{bmatrix} \eta_1 \mathbf{1}^T \\ \eta_2 \mathbf{1}^T \\ \cdots \\ \eta_K \mathbf{1}^T \end{bmatrix},$$

*where $\mathbf{1}$ denotes $n$ dimensional vectors with 1s, and $\epsilon_1, \ldots, \epsilon_K, \eta_1, \ldots, \eta_K$ are constants. Suppose the perturbed cost matrix is defined as $\tilde{C} = C + E + F$, then $\pi^*$ is also an optimal transport map with respect to the perturbed cost matrix $\tilde{C}$.*

Now we provide the proof of Thoerem 4.2.

*Proof.* Given
$$C_{ij} = -\log P_\theta(Y = j | X = x_i) = -\log P_s(Y = j | X = x_i),$$
the posteriors in the target distribution can be defined as $C_{ij}^* = -\log P_t(Y = j | X = x_i)$. From

$$P_t(Y = j | X = x_i) = P_s(Y = j | X = x_i) \frac{P_s(X = x_i) P_t(Y = j)}{P_t(X = x_i) P_s(Y = j)},$$

we can see that

$$
\begin{aligned}
C_{ij}^* &= -\log P_t(Y = j | X = x_i) \\
&= -\log P_s(Y = j | X = x_i) \frac{P_s(X = x_i) P_t(Y = j)}{P_t(X = x_i) P_s(Y = j)} \\
&= -\log P_s(Y = j | X = x_i) + \log P_s(Y = j) \\
&\quad - \log P_t(Y = j) - \log P_s(X = x_i) + \log P_t(X = x_i) \\
&= C_{ij} + E_{\cdot j} + F_{i\cdot}.
\end{aligned}
$$

where $E_{\cdot j} = \log P_s(Y = j) - \log P_t(Y = j)$, $F_{i\cdot} = -\log P_s(X = x_i) + \log P_t(X = x_i)$. And, assuming $\nu_j = \frac{1}{n} \sum_{i=1}^{n} \mathbb{1}[\hat{y}_i^* = j]$, where $\hat{y}^*$ is the Bayes classifier prediction such that

$$
\begin{aligned}
\hat{y}_i^* &= \arg\max_{j \in [K]} P_t(Y = j | X = x_i) \\
&= \arg\min_{j \in [K]} -\log P_t(Y = j | X = x_i).
\end{aligned}
$$

Then, optimal transport solution

$$\pi^* = \arg\min_{\gamma \in \Pi(\mu,\nu)} \langle \gamma, C^* \rangle$$

leads to Bayes classifier predictions by Theorem 4.1.

Finally, by Corollary D.2, we have

$$\pi^* = \arg\min_{\gamma \in \Pi(\mu,\nu)} \langle \gamma, C^* \rangle = \arg\min_{\gamma \in \Pi(\mu,\nu)} \langle \gamma, C \rangle.$$

$\square$

### D.3 Proof of Theorem 4.3

The proof of Theorem 4.3 relies on the following result of [52].

**Lemma D.3** ([52], Corollary 3.1.). *Let the primal linear programming problem be*

$$\min_{z}\{p^T z | Gz \geqslant g, z \geqslant 0\}$$

*and its dual be*

$$\max_{w}\{w^T g | w^T G \leqslant p^T, w \geqslant 0\}.$$

*Let $\bar{z}, \bar{w}$ be the primal, dual solution. And, let the perturbed primal linear programming problem be*

$$\min_{z}\{\hat{p}^T z | \hat{G}z \geqslant \hat{g}, z \geqslant 0\}$$

*and its dual be*

$$\max_{w}\{w^T \hat{g} | w^T \hat{G} \leqslant \hat{p}^T, w \geqslant 0\}.$$

*Let $\hat{z}, \hat{w}$ be the corresponding primal, dual solution.*

*Suppose that the primal and dual problems are solvable. Then,*

$$\left\| \begin{pmatrix} \bar{z} \\ \bar{w} \end{pmatrix} - \begin{pmatrix} \hat{z} \\ \hat{w} \end{pmatrix} \right\|_2 \leq \kappa \left\| \begin{matrix} [(G - \hat{G})\hat{z} - (g - \hat{g})]^- \\ [(G - \hat{G})^T \hat{w} - (p - \hat{p})]^+ \\ (p - \hat{p})\hat{z} - (g - \hat{g})\hat{w} \end{matrix} \right\|_p,$$

*where $1 \leq p \leq \infty$ and $\kappa$ is the Hoffmann constant determined by $p, G, g$. [30].*

This Lemma provides a bound for error in the solution of the perturbed linear program. Since discrete optimal transport can be translated to standard linear program, we obtain Theorem 4.3 by plugging in the definitions.

**Proof of Theorem 4.3** A discrete optimal transport problem

$$\min \left\{ \sum_{i,j} C_{i,j} \pi_{i,j} | \pi \mathbf{1} = \mu, \pi^T \mathbf{1} = \nu, \pi_{ij} \geq 0 \right\}$$

can be written as a linear program

$$\min\{p^T z | Gz \geqslant g, z \geqslant 0\},$$

where $p = vec(C), z = vec(\pi), H = \begin{bmatrix} \mathbf{1}_n^T \otimes \mathbb{I}_K \\ \mathbb{I}_n \otimes \mathbf{1}_K^T \end{bmatrix}, G = \begin{bmatrix} H \\ -H \end{bmatrix}, g = \begin{bmatrix} \mu \\ \nu \\ -\mu \\ -\nu \end{bmatrix}$. Note that the equality

constraints are converted to stacked inequalities. We have noisy cost matrix and label distribution in our setting, which leads to the perturbation on cost matrix $C$ and $\nu$ such that the perturbed cost matrix is $\hat{C} = C + \Delta_C$, the perturbed label distribution $\hat{\nu} = \nu + \Delta_\nu$, such that $\hat{g} = g + \Delta_g$ where

$\Delta_g = \begin{bmatrix} 0 \\ \hat{\nu} - \nu \\ 0 \\ -\hat{\nu} + \nu \end{bmatrix}$. Since we don't have perturbation on the constraint matrix $G, \hat{G} = G$. By plugging

in these terms to Lemma D.3.

$$\left\| \begin{pmatrix} \bar{z} \\ \bar{w} \end{pmatrix} - \begin{pmatrix} \hat{z} \\ \hat{w} \end{pmatrix} \right\|_2 \le \kappa \left\| \begin{matrix} [(G - \hat{G})\hat{z} - (g - \hat{g})]_- \\ [(G - \hat{G})^T \hat{w} - (p - \hat{p})]_+ \\ (p - \hat{p})\hat{z} - (g - \hat{g})\hat{w} \end{matrix} \right\|_2$$

$$= \kappa \left\| \begin{matrix} [g - \hat{g}]_- \\ [p - \hat{p}]_+ \\ (p - \hat{p})\hat{z} - (g - \hat{g})\hat{w} \end{matrix} \right\|_2$$

$$= \kappa \left\| \begin{matrix} 0 \\ [\hat{\nu} - \nu]_- \\ 0 \\ [\nu - \hat{\nu}]_- \\ [p - \hat{p}]_+ \\ p\hat{z} - g\hat{w} \end{matrix} \right\|_2 \qquad \because \text{ Optimality of } \hat{z}, \hat{w} \text{ in the perturbed problem.}$$

$$= \kappa \left\| \begin{matrix} [\hat{\nu} - \nu] \\ [p - \hat{p}]_+ \\ p\hat{z} - g\hat{w} \end{matrix} \right\|_2$$

Then, we have

$$\left\| \begin{pmatrix} \bar{z} \\ \bar{w} \end{pmatrix} - \begin{pmatrix} \hat{z} \\ \hat{w} \end{pmatrix} \right\|_2^2 \le \kappa^2 \left\| \begin{matrix} [\hat{\nu} - \nu] \\ [p - \hat{p}]_+ \\ p\hat{z} - g\hat{w} \end{matrix} \right\|_2^2 .$$

Then,

$$\| \bar{z} - \hat{z} \|_2^2 \le \kappa^2 \left\| \begin{matrix} [\hat{\nu} - \nu] \\ [p - \hat{p}]_+ \\ p\hat{z} - g\hat{w} \end{matrix} \right\|_2^2 - \| \bar{w} - \hat{w} \|_2^2$$

Plugging in $\bar{z} = vec(\pi), \hat{z} = vec(\hat{\pi}), \Delta_\nu = \hat{\nu} - \nu, \Delta_C = \hat{C} - C$, and rearranging, we obtain

$$\| \pi - \hat{\pi} \|_F^2 \le \kappa^2 \left( \| \Delta_\nu \|_2^2 + \| [vec(\Delta_C)]_+ \|_2^2 + \| vec(C)^T vec(\hat{\pi}) - g^T \hat{w} \|_2^2 \right) - \| w - \hat{w} \|_2^2 .$$

We use the definition of the Hoffman constant $\kappa$ from [52]. Computing Hoffman constant or even bounding it has been a long-standing problem [5; 45; 46]. However, it has been shown that the Hoffman constant is a finite real number [52], and specifically under our problem setup, it is independent from perturbations and only related to original optimization problem. This suggests the possibility to regularize the parameters $C, G, g$ in the original problem such that $\kappa$ does not depend on the dimensionality of cost matrix or target distribution. We leave further exploration of $\kappa$ in this context as future work.

### D.4 Bounding label distribution estimation errors in few-shot learning

In few-shot learning, we assume that a few labeled samples per class are given. They can be used for estimating label distribution in the target distribution using label shift estimation methods [35; 6; 24]. They comes with the sample complexity analysis under the label shift assumptions, which can be used to obtain bound the label distribution estimation errors.

**Lemma D.4.** *Let $m$ and $n$ denote the number of few-shot learning data and test datapoints, $w_i = \nu_i / \nu_i^s$ and $\hat{w}_i = \hat{\nu}_i / \nu_i^s$. Also let $\sigma_{\min}$ be the smallest eigenvalue of the covariance matrix $V_{\hat{y},y}$ where $[V_{\hat{y},y}]_{i,j} = P_s(f(x) = i, y = j)$. For $m > 80 \log(m) \sigma_{\min}^{-2}$ and constant $c > 0$, the perturbation $\Delta_\nu$ may be bounded as*

$$\| \Delta_\nu \|^2 \le \| \nu^s \|^2 \frac{c}{\sigma_{\min}^2} \left( \| w \|^2 \frac{\log m}{m} + K \frac{\log n}{n} \right),$$

*with probability at least $1 - 3Km^{-10} - 2Kn^{-10}$.*

The proof of Lemma D.4 relies on the following result of [34].

**Lemma D.5.** *Assume that*

*1. $\forall x \in \mathcal{X}$ and $\forall y \in \mathcal{Y}$, $P_s(x|y) = P_t(x|y)$,*

*2. if $P_t(y) > 0$ then $P_s(y) > 0$ $\forall y \in \mathcal{Y}$, and*

3. *the expected confusion matrix $C_s(f) = P_s(f(x), y) \in \mathbb{R}^{K \times K}$ for classifier $f$ is invertible.*

*Then, there exists a constant $c > 0$ such that for all $m > 80 \log(m)\sigma_{\min}^{-2}$, with probability at least $1 - 3Km^{-10} - 2Kn^{-10}$,*

$$||\hat{w} - w||^2 \leq \frac{c}{\sigma_{\min}^2} \left( ||w||^2 \frac{\log m}{m} + K \frac{\log n}{n} \right).$$

*Proof of Lemma D.4.* Where all norms are Euclidean unless otherwise denoted, we have that

$$||\Delta_\nu||^2 = ||\hat{\nu} - \nu||^2$$
$$= ||\nu^s \otimes (\hat{w} - w)||^2$$

for element-wise multiplication operation $\otimes$. Further,

$$||\nu^s \otimes (\hat{w} - w)||^2 \leq ||\nu^s(\hat{w} - w)||_F^2$$
$$\leq ||\nu^s||_2^2 ||\hat{w} - w||^2$$
$$\leq ||\nu^s||^2 \frac{c}{\sigma_{\min}^2} \left( ||w||^2 \frac{\log m}{m} + K \frac{\log n}{n} \right),$$

where the last line follows from Lemma D.5 with probability at least $1 - 3Km^{-10} - 2Kn^{-10}$. $\quad\square$

## D.5 Bounding the perturbation on cost matrix

Further, we can bound $[vec(\Delta_C)]_+$ by the Total Variation Distance (TVD) between $P_s$ and $P_\theta$ as follows.

**Lemma D.6.** *Let $\tau = \frac{1}{2}||P_s - P_\theta||_1$ denote the Total Variation Distance between $P_s$ and $P_\theta$ and define $\min(C) = \min_{i,j} C_{ij}$. Then,*

$$||[vec(\Delta_C)]_+|| \leq \sqrt{m(K-1)} \log \left( \frac{\tau}{\min(C)} + 1 \right).$$

*Proof of Lemma D.6.* For each element $\Delta_C^{(ij)}$ of $\Delta_C$, we have that

$$\Delta_C^{(ij)} = \log P_\theta(Y = j | X = x_i) - \log P_s(Y = j | X = x_i)$$
$$= \log \frac{P_\theta(Y = j | X = x_i)}{P_s(Y = j | X = x_i)}.$$

For $i, j$ such that $P_\theta(Y = j | X = x_i) \leq P_s(Y = j | X = x_i)$, clearly $\Delta_C^{(ij)} \leq 0$ and so $[vec(\Delta_C^{(ij)})]_+ = 0$.

Otherwise, for $i, j$ such that $P_\theta(Y = j | X = x_i) > P_s(Y = j | X = x_i)$, it follows that $\Delta_C^{(ij)} > 0$ and

$$\Delta_C^{(ij)} = \log \frac{P_\theta(Y = j | X = x_i) - P_s(Y = j | X = x_i) + P_s(Y = j | X = x_i)}{P_s(Y = j | X = x_i)}$$
$$\leq \log \left( \frac{\tau}{P_s(Y = j | X = x_i)} + 1 \right)$$
$$\leq \log \left( \frac{\tau}{\min(C)} + 1 \right).$$

For each $i \in [m]$, there are at most $K - 1$ possible $j$ such that $P_\theta(Y = j | X = x_i) > P_s(Y = j | X = x_i)$, because $\sum_{j \in [K]} P_\theta(Y = j | X = x_i) = \sum_{j \in K} P_s(Y = j | X = x_i)$. Therefore, there are at most $m(K-1)$ pairs $(i, j) \in [m] \times [K]$ such that $0 < \Delta_C^{(ij)} \leq \log \left( \frac{\tau}{\min(C)} + 1 \right)$. Thus,

$$||[vec(\Delta_C)]_+|| \leq \sqrt{m(K-1)} \log \left( \frac{\tau}{\min(C)} + 1 \right).$$

$\square$

## D.6 Proof of Theorem 4.4

We provide the proof of Theorem 4.4.

*Proof.* By assumption, we have $P_\theta(Y|X) = P_s(Y|X)$. By the result of Theorem 4.2, $y_{\text{OTTER}}$ samples are generated by $\arg\max_{j\in[K]} P_t(Y = j|X = x)$. Suppose $r^* = P_t(Y)/P_s(Y)$. Then,

$$
\begin{aligned}
P_{\theta,r^*}(Y = j|X = x) &= \frac{r_j^* P_\theta(Y = j|X = x)}{\sum_{j'=1}^{K} r_{j'}^* P_\theta(Y = j'|X = x)} \\
&= \frac{r_j^* P_s(Y = j|X = x)}{\sum_{j'=1}^{K} r_{j'}^* P_s(Y = j'|X = x)} \\
&= \frac{\frac{P_t(Y=j)}{P_s(Y=j)} \frac{P_s(X=x|Y=j)P_s(Y=j)}{P_s(X=x)}}{\sum_{j'=1}^{K} \frac{P_t(Y=j')}{P_s(Y=j')} \frac{P_s(X=x|Y=j')P_s(Y=j')}{P_s(X=x)}} \\
&= \frac{P_t(Y = j)P_s(X = x|Y = j)}{\sum_{j'=1}^{K} P_t(Y = j')P_s(X = x|Y = j')} \\
&= \frac{P_t(Y = j)P_t(X = x|Y = j)}{\sum_{j'=1}^{K} P_t(Y = j')P_t(X = x|Y = j')} \\
&= \frac{\frac{P_t(Y=j)P_t(Y=j|X=x)P_t(X=x)}{P_t(Y=j)}}{\sum_{j'=1}^{K} \frac{P_t(Y=j')P_t(Y=j'|X=x)P_t(X=x)}{P_t(Y=j')}} \\
&= \frac{P_t(Y = j|X = x)}{\sum_{j'=1}^{K} P_t(Y = j'|X = x)}
\end{aligned}
$$

Thus, we have

$$
\begin{aligned}
y_{\text{R-OTTER}} &= \arg\max_j P_{\theta,r^*}(Y = j|X = x) \\
&= \arg\max_j \frac{P_t(Y = j|X = x)}{\sum_{j'=1}^{K} P_t(Y = j'|X = x)} \\
&= \arg\max_j P_t(Y = j|X = x) \\
&= y_{\text{OTTER}},
\end{aligned}
$$

which implies $r^*$ is an optimal parameter. $\square$

|  | CIFAR10 | CIFAR100 | Caltech101 | Caltech256 | Food101 | STL10 |
|---|---|---|---|---|---|---|
| n | 10,000 | 10,000 | 7,162 | 22,897 | 25,250 | 8,000 |
| K | 10 | 100 | 101 | 256 | 101 | 10 |
| Imbalance | 1.00 | 1.00 | 49.06 | 15.94 | 1.00 | 1.00 |

|  | SUN397 | Flowers102 | EuroSAT | Oxford-IIIT-Pet | STANFORD-Cars | Country211 |
|---|---|---|---|---|---|---|
| n | 87,004 | 6,149 | 22,000 | 3,669 | 8,041 | 211,00 |
| K | 397 | 102 | 10 | 37 | 196 | 211 |
| Imbalance | 25.43 | 11.90 | 1.67 | 1.14 | 2.83 | 1.00 |

|  | DTD | CUB | ImageNet | ImageNet-r | ImageNet-Sketch |
|---|---|---|---|---|---|
| n | 1,880 | 5,794 | 40,000 | 26,000 | 40,889 |
| K | 47 | 200 | 1,000 | 200 | 1,000 |
| Imbalance | 1.00 | 2.73 | 1.00 | 13.23 | 1.03 |

|  | Amazon | Gender | CivilComments | HateXplain |
|---|---|---|---|---|
| n | 89,078 | 21,750 | 131,782 | 1,621 |
| K | 2 | 2 | 2 | 3 |
| Imbalance | 19.45 | 6.03 | 8.26 | 1.52 |

Table 7: Statistics of the test dataset in each task. Class imbalance is measured by $\dfrac{\max_{j \in [K]} P_t(Y = j)}{\min_{j \in [K]} P_t(Y = j)}$.

# E  Experiment details

## E.1  Datasets

**Zeroshot image classification datasets**  We use CIFAR10, CIFAR100 [33], Caltech101 [22], Caltech256 [25], Food101 [8], STL10 [16], SUN397 [67], Flower102 [42], EuroSAT [27], Oxford-IIIT Pet [44], Stanford Cars [32], DTD [14], CUB [61], ImageNet [18], ImageNet-r [29], and ImageNet-Sketch [63].

**Zeroshot text classification datasets**  We use Amazon [41], Gender [20], CivilComments [7], and HateXplain [39].

| | | CIFAR10 | CIFAR100 | Caltech101 | Caltech256 | Country211 | DTD | EUROSAT | Flower102 |
|---|---|---|---|---|---|---|---|---|---|
| **RN50** | Zero-shot | 66.5 | 38.7 | 73.5 | 73.0 | 13.3 | 37.5 | 18.2 | 57.4 |
| | CLIPPR | 69.2 | 20.1 | 59.4 | 67.1 | 8.2 | 13.0 | 19.3 | 30.7 |
| | Prior Matching | 76.4(±0) | 41.1(±2.5) | 44.3(±9.7) | 9.8(±1.6) | 13.11(±0.6) | 41.6(±0.2) | **31.5 (±6.0)** | 50.0(±2.8) |
| | OTTER(Ours) | **76.8** | **44.4** | **83.7** | **80.5** | **14.1** | **42.2** | 26.8 | **64.3** |
| **RN101** | Zero-shot | 79.1 | 46.3 | 81.0 | 76.6 | 14.8 | 37.3 | 33.8 | 61.3 |
| | CLIPPR | 80.4 | 24.1 | 52.5 | 65.4 | 7.6 | 5.9 | **37.1** | 30.5 |
| | Prior Matching | 80.3 (±0.1) | 46.7 (±1.3) | 57.6 (±14.2) | 9.6 (±2.4) | 15.1 (±0.7) | 39.4 (±0.1) | 34.0 (±5.9) | 49. (±20.9) |
| | OTTER(Ours) | **81.1** | **50.9** | **89.4** | **83.7** | **16.0** | **40.7** | 33.0 | **67.7** |
| **ViT-B/32** | Zero-shot | 88.9 | 58.5 | 80.3 | 78.2 | 15.5 | 40.7 | 29.7 | 59.3 |
| | CLIPPR | **89.9** | 37.8 | 55.5 | 69.3 | 8.5 | 8.4 | 31.8 | 29.8 |
| | Prior Matching | 89.7 (±0) | 58.1 (±1.2) | 54.7 (±14.1) | 9.5 (±1.6) | 14.8 (±0.1) | 42.9 (±0.1) | 41.2 (±0.7) | 51.5 (±3.2) |
| | OTTER(Ours) | 89.7 | **64.4** | **88.0** | **85.0** | **15.9** | **45.1** | **44.9** | **68.1** |
| **ViT-B/16** | Zero-shot | 88.3 | 63.8 | 79.8 | 79.8 | 19.8 | 39.0 | 32.9 | 64.0 |
| | CLIPPR | 90.3 | 41.6 | 60.3 | 75.4 | 12.0 | 9.1 | 36.4 | 36.2 |
| | Prior Matching | 91.3 (±0) | 64.1 (±2.7) | 59.3 (±15.4) | 9.5 (±1.5) | 19.0 (±0.1) | 42.1 (±0.1) | 41.6 (±0.8) | 54.0 (±14.1) |
| | OTTER(Ours) | **91.7** | **67.9** | **88.7** | **87.0** | **21.1** | **44.4** | 42.0 | **70.8** |
| **ViT-L/14** | Zero-shot | 95.0 | 72.3 | 78.2 | 83.4 | 28.2 | 50.8 | 25.8 | 72.3 |
| | CLIPPR | **96.2** | 57.9 | 66.8 | 80.2 | 17.4 | 12.8 | 31.5 | 44.3 |
| | Prior Matching | 95.2 (±0) | 73.5 (±2.5) | 68.5 (±19.7) | 9.4 (±1.2) | 27.5 (±0.3) | **51.5 (±0)** | **59.2 (±0.4)** | 66.5 (±17.9) |
| | OTTER(Ours) | 96.0 | **77.7** | **91.7** | **90.7** | **29.5** | 51.0 | 57.6 | **81.4** |

Table 8: CLIP Zero-shot image classification accuracy (%)

### E.2 Zero-shot classification setup

**Image zero-shot classification** For zero-shot image classification, we emply CLIP [49] models. We used "a photo of a [CLASS]" prompt. Scores are computed by $s_\theta(x_i, j) = P_\theta(Y = j | X = x_i) = \dfrac{\exp\left(\cos(f(x_i), g(y_j))/\tau\right)}{\sum_{j'=1}^{K} \exp\left(\cos(f(x_i), g(y_{j'}))/\tau\right)}$ for image $x_i$ regarding the label $j$, given the image encoder $f$, the text encoder $g$. Cost matrix is constructed by $C = [C_{ij}]_{i \in [n], j \in [K]}$, where $c_{ij} = -\log s_\theta(x_i, j)$. We run Algorithm 1 with the true class balance of the test dataset.

**Text zero-shot classification** We employ BERT and text-embedding-ada-002 sentence embeddings for text classification [50]. This process parallels the methodology used in image zero-shot classification — we compute the prediction scores from the cosine similarity and then construct the cost matrix with the negative log probabilities.

### E.3 Detailed experiment results of Section 5.1

**Ablation on zero-shot models** For the ablation study on zero-shot models, we provide experimental results with varying architectures, given the exact prior. Table 8, 9 show the image zero-shot classification results, and Table 10 shows the text zero-shot classification results. We also provide another baseline results with CLIPPR [31], which uses the label distribution for adapter training. CLIPPR is similar with Prior Matching in the point that it requires adapter training, but it has more adapter layers and additional loss function to make the predictions stick to the original prediction scores. While the performance gain varies, we can observe that OTTER is effective for the most cases.

| | | Food101 | Pet | Stanford | STL | SUN397 | CUB | ImageNet | ImageNet-r | Imagenet-Sketch |
|---|---|---|---|---|---|---|---|---|---|---|
| RN50 | Zero-shot | 76.3 | 80.5 | 45.6 | 93.1 | 42.4 | 39.8 | 51.5 | 35.0 | 5.3 |
| | CLIPPR | 69.5 | 61.9 | 32.5 | 94.4 | 45.0 | 19.0 | 31.1 | 23.8 | 0.7 |
| | Prior Matching | 77.5 (±1.4) | 77.4 (±0.3) | 27.5 (±1.9) | 94.7 (±0) | 12.3 (±3.9) | 36. (±0.1) | 44.6 (±0.1) | 7.8 (±1.8) | 5.0 (±0.0) |
| | OTTER(Ours) | **81.1** | **82.9** | **49.2** | **95.5** | **48.1** | **42.7** | **54.0** | **37.6** | **5.8** |
| RN101 | Zero-shot | 80.9 | 80.2 | 52.8 | 96.5 | 36.9 | 34.9 | 53.4 | 41.9 | 6.5 |
| | CLIPPR | 72.3 | 57.7 | 31.9 | 97.2 | 40.2 | 9.6 | 25.7 | 26.9 | 0.8 |
| | Prior Matching | 80.8 (±3.7) | 77.2 (±0.4) | 34.8 (±2.9) | 96.7 (±6) | 17.6 (±5) | 33.6 (±0.1) | 46.7 (±0.1) | 9.5 (±3.1) | 6.5 (±0.1) |
| | OTTER(Ours) | **84.4** | **84.2** | **55.2** | 97.2 | **44.6** | **41.0** | **56.0** | **44.5** | **7.3** |
| ViT-B/32 | Zero-shot | 80.2 | 81.7 | 49.1 | 97.1 | 39.1 | 39.9 | 55.6 | 61.0 | 34.2 |
| | CLIPPR | 73.9 | 58.0 | 30.6 | 97.5 | 41.6 | 11.9 | 32.6 | 51.7 | 19.9 |
| | Prior Matching | 80.0 (±4.4) | 77.5 (±0.6) | 31.5 (±1.1) | 97.4 (±0.0) | 11.3 (±3.1) | 36.3 (±0.9) | 48.3 (±0.1) | 11.7 (±0.8) | 30.4 (±0.3) |
| | OTTER(Ours) | **85.0** | **86.2** | **52.1** | **97.8** | **44.7** | **45.4** | **57.7** | **64.7** | **39.3** |
| ViT-B/16 | Zero-shot | 85.6 | 83.8 | 55.7 | 98.0 | 47.1 | 46.0 | 60.2 | 68.9 | 39.8 |
| | CLIPPR | 80.5 | 64.5 | 40.7 | 98.6 | 50.4 | 20.5 | 37.7 | 59.8 | 25.6 |
| | Prior Matching | 86.8 (±3.1) | 82.0 (±0.3) | 39.8 (±2.6) | 98.4 (±0) | 6.7 (±1.6) | 40.4 (±0) | 53.6 (±0.1) | 16.6 (±3.5) | 36.5 (±0.4) |
| | OTTER(Ours) | **89.9** | **88.8** | **59.7** | 98.6 | **54.1** | **50.4** | **62.9** | **72.4** | **44.5** |
| ViT-L/14 | Zero-shot | 89.8 | 87.9 | 64.1 | 99.2 | 64.9 | 47.3 | 67.5 | 80.9 | 51.9 |
| | CLIPPR | 87.9 | 82.7 | 54.0 | 99.6 | 68.2 | 27.3 | 45.4 | 74.5 | 35.5 |
| | Prior Matching | 90.4 (±3.8) | 84.0 (±0.5) | 53.0 (±15.6) | 99.3 (±0) | 26.5 (±1.1) | 43.9 (±0.3) | 62.3 (±0.1) | 17.6 (±5.1) | 47.2 (±0.4) |
| | OTTER(Ours) | **93.6** | **91.0** | **70.0** | 99.4 | **71.5** | **55.4** | **70.2** | **83.7** | **55.2** |

Table 9: CLIP Zero-shot image classification accuracy (%) continued.

| | | Amazon review | GenderBias | CivilComments | HateXplain |
|---|---|---|---|---|---|
| BERT | Zero-shot | 74.0 | 84.1 | 48.4 | 30.9 |
| | CLIPPR | 74.9 | 84.2 | 49.4 | 31.0 |
| | Prior Matching | 58.8 (± 46.4) | 41.4 (± 39.6) | 57.2 (± 37.7) | 31.3 (±3.3) |
| | OTTER(Ours) | **91.7** | **91.9** | **81.4** | **34.3** |
| Ada | Zero-shot | 72.3 | 50.9 | 56.2 | 27.9 |
| | CLIPPR | 73.7 | 53.0 | 57.6 | 27.6 |
| | Prior Matching | 58.8 (± 43.7) | 50.0 (±41.3) | 55.2 (±35.5) | 32.4 (± 4.0) |
| | OTTER(Ours) | **97.0** | **73.7** | **82.0** | 32.0 |

Table 10: Text embedding zero-shot classification mean accuracy and standard deviation (%)

**Ablation on the class balance specification**  We conducted a semi-synthetic experiment to investigate the sensitivity to the label distribution specification error in real datasets. We generate the noisy label distribution specification and see how the accuracy changes. We control the noise in the label distribution specification as follows. Given the true class balance $\nu^*$, first we make adversarial class balance $\nu^{adv}$ such that $\nu^{adv}_{j^*} = 1$ for $j^* = \arg\min_{j \in [K]} \nu^*_j$ and $\nu^{adv}_j = 0$ for $j \neq j^*$. To measure distance between class balance specification and true class balance, we use the total variance $TV(\nu, \hat{\nu}) = \frac{1}{2} ||\nu - \hat{\nu}||_1$. Next, we intepolate $\nu^*$ and $\nu^{adv}$ such that $TV(\nu^*, \nu^\alpha) = \alpha$, by $\nu^\alpha = (1 - \frac{\alpha}{TV(\nu^*, \nu^{adv})})\nu^* + \frac{\alpha}{TV(\nu^*, \nu^{adv})}\nu^{adv}$. We set the interval of alpha as 0.01 and vary it up to 0.2.

Figure 5 shows the result. We observe the sensitivity to the label distribution specification error varies depending on the datasets, but generally we can observe that the accuracy degrades linearly proportionally to the class balance error. While the result may vary depending on the interaction between class balance error and calibration error in cost matrix, we can expect performance improvement if the class balance specification is good enough.

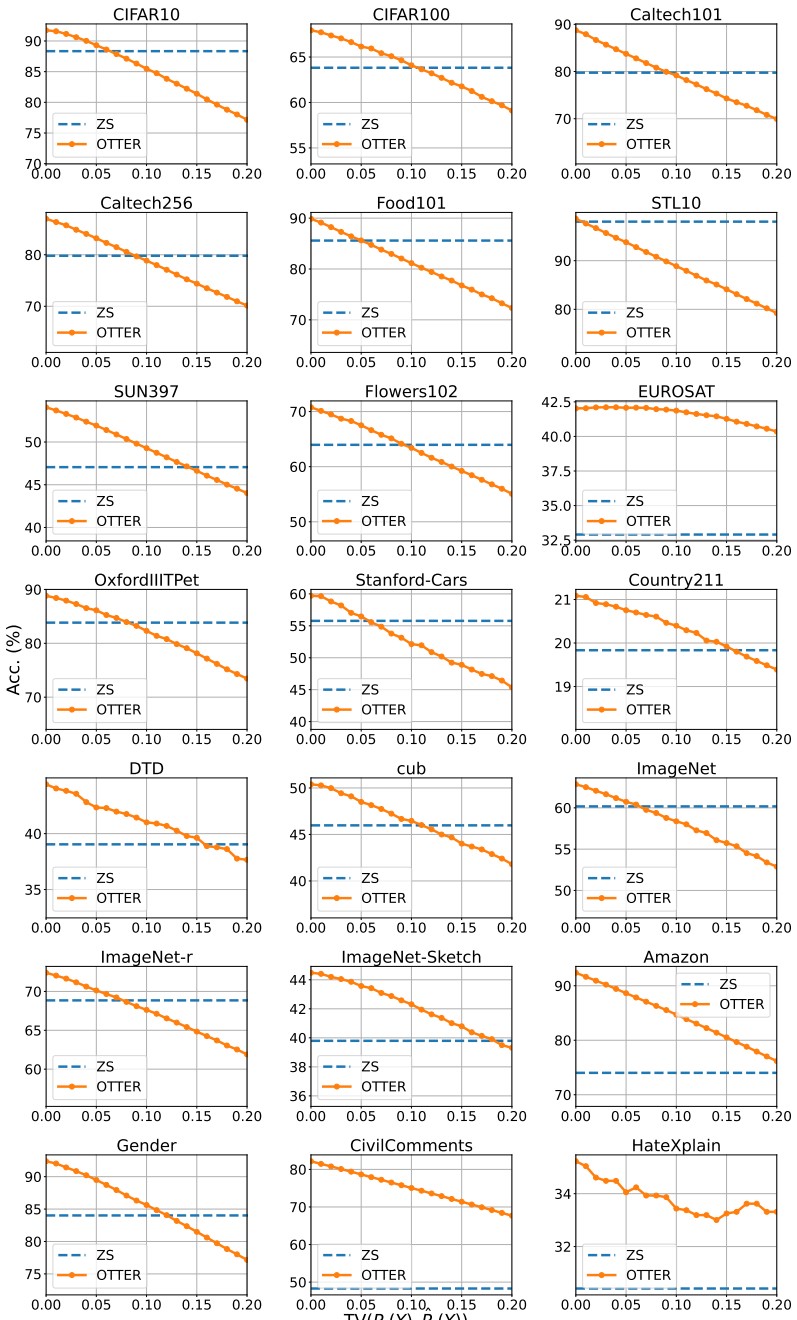

Figure 5: Ablation experiment on the class balance specification. X-axis represents the total variation distance between the class specification true class balance $P_t(Y)$ and $\hat{P}_t(Y)$. Y-axis represents accuracy. ViT-B/16 is used as the image zero-shot classifier, and BERT is used as the text zero-shot classifier.

**Inference time comparison** To show that the additional computation complexity induced by OTTER is not heavy, we measured the time consumption (in seconds) for the inference step in the experiments in Section 5.1, with the pre-computed embeddings. Table 11 presents the result. Time reduction column represents the time reduction rate of OTTER compared to PM. Measurements were

| Dataset | n | ZS | PM | OTTER | Time reudction (%) |
|---|---|---|---|---|---|
| CIFAR10 | 10000 | 0.0381 | 3.7127 | 0.0733 | 98.03 |
| CIFAR100 | 10000 | 0.0462 | 3.6296 | 0.1947 | 94.64 |
| Caltech101 | 7162 | 0.0298 | 3.6445 | 0.1188 | 96.74 |
| Caltech256 | 22897 | 0.2111 | 3.9597 | 0.8568 | 78.36 |
| Food101 | 25250 | 0.1186 | 3.6968 | 0.4969 | 86.56 |
| STL10 | 8000 | 0.0304 | 3.4877 | 0.0546 | 98.43 |
| SUN397 | 87004 | 1.1233 | 33.0386 | 10.5316 | 68.12 |
| Flowers102 | 6149 | 0.0280 | 3.7216 | 0.0959 | 97.42 |
| EuroSAT | 22000 | 0.0826 | 3.6655 | 0.3491 | 90.48 |
| OXFORD-IIIT-Pet | 3669 | 0.0137 | 3.3901 | 0.0233 | 99.31 |
| STANFORD-Cars | 8041 | 0.0413 | 3.4910 | 0.1964 | 94.37 |
| Country211 | 21100 | 0.1285 | 3.7665 | 1.0537 | 72.02 |
| DTD | 1880 | 0.0070 | 3.4603 | 0.0156 | 99.55 |
| CUB | 5794 | 0.0306 | 3.5583 | 0.1410 | 96.04 |
| ImageNet | 40000 | 0.9954 | 37.6932 | 8.1003 | 78.51 |
| ImageNet-r | 26000 | 0.1921 | 3.8331 | 0.9834 | 74.35 |
| ImageNet-Sketch | 40889 | 1.0189 | 38.4853 | 9.0579 | 76.46 |

Table 11: Inference time comparison with pre-computed embeddings (in seconds).

| | ZS | PM | OT | ZS + CD | PM + CD | OT + CD |
|---|---|---|---|---|---|---|
| EuroSAT | 32.90 | 11.36 | 42.03 | 53.62 | 11.37 | **57.15** |
| Oxford-IIIT-Pet | 83.84 | 23.11 | 88.83 | 87.95 | 16.33 | **91.01** |
| DTD | 39.04 | 8.83 | **44.41** | 42.87 | 14.73 | 43.24 |
| CUB | 45.98 | 10.34 | 50.40 | 55.51 | 11.49 | **58.47** |
| ImageNet | 60.18 | 12.42 | 62.86 | 66.46 | 14.08 | **68.05** |

Table 12: Accuracy in the prompt-enhanced zero-shot classification by Classification by Description (CD) [40]. We can observe OTTER's capability to provide further enhancements upon refined the improvements achieved through refined prompts.

taken using a machine equipped with an Intel® Core™ i7-11700K @ 3.60GHz processor, 64GB RAM, and NVIDIA GPU RTX-4090. For most cases ($n < 30000$), our method takes less than 1 second, while the prior matching baseline takes more than 3 seconds. It's worth noting that the time consumption for computing embeddings is more substantial; even 10 seconds is negligible compared to the embedding time consumption (ranging from 5 to 30 minutes for each evaluation set), which is common for all inference conditions.

**Ablation on prompts** Recent studies have demonstrated the efficacy of enhancing prompts as a means to improve zero-shot models [70; 40]. In order to further illustrate the potential enhancements offered by OTTER beyond enhanced prompts, we reproduced Menon and Vondrick [40]'s approach (Classification by Description, CD), which employs multiple prompts generated by language models and takes max scores of them for each class. We applied OTTER to CD. The results of this experiment are summarized in Table 12. As anticipated, OTTER exhibits enhancements in zero-shot classification, even when prompt sets are refined using language models.

| Dataset | $TV(\nu^*, \hat{\nu}^{zs})$ | $TV(\nu^*, \hat{\nu}^{lp})$ | Dataset | $TV(\nu^*, \hat{\nu}^{zs})$ | $TV(\nu^*, \hat{\nu}^{lp})$ |
|---|---|---|---|---|---|
| CIFAR10 | 0.071 | **0.038** | STL10 | 0.021 | **0.011** |
| CIFAR100 | 0.219 | **0.153** | SUN397 | 0.503 | **0.458** |
| Caltech101 | 0.130 | **0.041** | CUB | 0.245 | **0.102** |
| Caltech256 | 0.126 | **0.081** | ImageNet | **0.175** | **0.175** |
| Country211 | 0.439 | **0.336** | ImageNet-r | 0.210 | **0.189** |
| DTD | 0.441 | **0.160** | ImageNet-sketch | 0.236 | **0.211** |
| EUROSAT | 0.404 | **0.084** | Amazon | **0.090** | 0.253 |
| Flowers102 | 0.202 | **0.067** | CivilComments | **0.369** | 0.383 |
| Food101 | 0.112 | **0.090** | Gender | **0.083** | 0.155 |
| Oxford-IIIT-Pet | 0.219 | **0.114** | HateXplain | 0.253 | **0.203** |
| Stanford-Cars | 0.255 | **0.143** | | | |

Table 13: Class balance estimation error with BBSE in Section 5.3. We report the mean of 10 different random samplings of the validation set. Lower is better.

### E.4 Detailed experiment results of Section 5.3

**Label distribution estimation errors** We report the label distribution estimation errors in Section 5.3. As a metric, we use the total variance $TV(\nu, \hat{\nu}) = \frac{1}{2} ||\nu - \hat{\nu}||_1$. We use zeroshot prediction scores and linear probing prediction scores for BBSE. $\hat{\nu}^{zs}$ denotes the estimated class balance based on zero-shot prediction scores, and $\hat{\nu}^{lp}$ represents the estimated class balance based on linear probing prediction scores.

Table 13 shows the result. We can see that total variation decreases with linear probing in image classification tasks since they reduces the violation of label shift assumptions. However, total variation increases in text classification tasks due to the small number of labeled sample size, following the size of label space ($K = 2$ or 3). Accordingly, we can expect OTTER will be more useful with linear probing, and just rebalancing zero-shot predictions with OTTER could be enough for text classification tasks.

**Ablation experiments on linear probing** We provide full results of Section 5.3. Specifically, we additionally report the results of combination with linear probing in text classification tasks and the results of zero-shot classification results in image classification tasks.

The results are presented in Table 2. While OTTER often provides additional improvement over LP, zero-shot classification was a strong baseline in image classification tasks. Meanwhile, class balance adaptation in text classification tasks is effective in all cases, giving a significant improvement over zero-shot predictions.

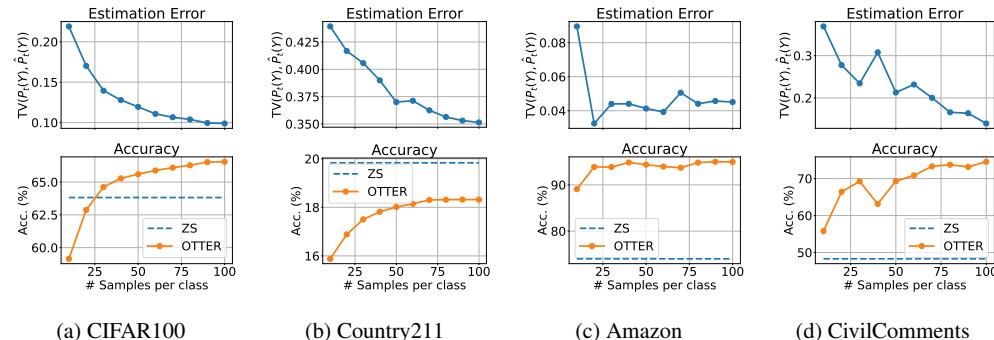

|   (a) CIFAR100   |   (b) Country211   |   (c) Amazon   |   (d) CivilComments   |

Figure 6: Ablation experiment on the number of samples. We report the mean of 10 different samplings in each setting. We use ViT-B/16 for image classification, and BERT for text classification.

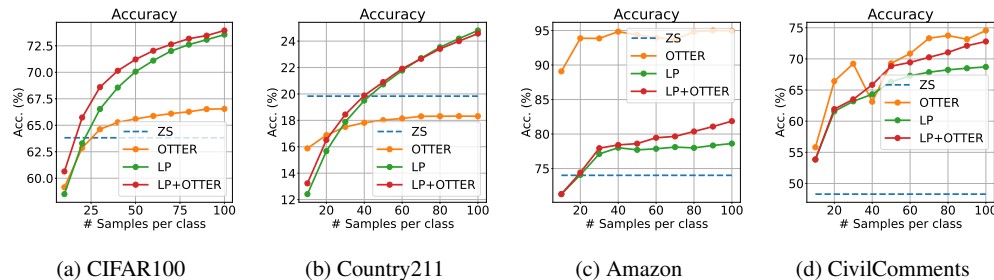

|   (a) CIFAR100   |   (b) Country211   |   (c) Amazon   |   (d) CivilComments   |

Figure 7: Comparison between OTTER, LP, and LP+OTTER with varying the number of samples. We report the mean of 10 different samplings in each setting. We use ViT-B/16 for image classification, and BERT for text classification.

**Ablation experiments on the number of examples per class**  Few-shot adaptation scenario assumes we have access to labeled data to estimate the target distribution. We hypothesize that an increase in the number of labeled samples enhances the accuracy of the class balance estimation, thereby improving the performance of OTTER. To test this hypothesis, we use few-shot adaptation in image and text classification tasks, without linear probing. The experiment varies the number of samples per class from 10 to 100, anticipating a reduction in class balance estimation error and an improvement in OTTER's accuracy with the increase in labeled samples.

The results, as depicted in Figure 6, corroborate our hypothesis. It is evident that the error in class balance estimation diminishes with an increasing number of samples, leading to a subsequent enhancement in the accuracy of OTTER.

**Comparison between OTTER and Linear Probing with varying number of classes**  In the few-shot adaptation scenario, we explored three approaches: OTTER, linear probing (LP), and a combination of LP + OTTER. We formulated two hypotheses. The first posits that OTTER might outperform LP, particularly in situations with a limited number of samples. The second hypothesis suggests that OTTER could provide further enhancements to LP even when LP already surpasses the naive version of OTTER. This experiment was conducted using the same setup as the previous one.

The results, displayed in Figure 7, reveal several insights regarding our hypotheses. To begin with, OTTER demonstrates performance on par with LP, especially in scenarios with fewer samples. Interestingly, OTTER achieves superior accuracy compared to LP in datasets like Amazon and CivilComments, characterized by a small number of classes ($K = 2$), resulting in a relatively low total sample count. Furthermore, it is observed that incorporating OTTER into LP leads to an average increase in accuracy.

### E.5   Detailed experiment setup and results of Section 5.4

**Class hierarchy**  We used the following superclasses and subclasses classes for the proof of concept.

|          | BBSE  | H-BBSE |
|----------|-------|--------|
| RN50     | 0.335 | **0.246** |
| RN101    | 0.378 | **0.294** |
| ViT-B/32 | **0.156** | 0.167 |
| ViT-B/16 | 0.287 | **0.246** |
| ViT-L/14 | **0.131** | 0.152 |

Table 14: Class balance estimation error in the Section 5.4 experiment. Class balance estimation error is measured by total variation distance. We report the mean of 10 different samplings of the validation set. H-BBSE denotes the class balance estimation using hierarchy upon BBSE.

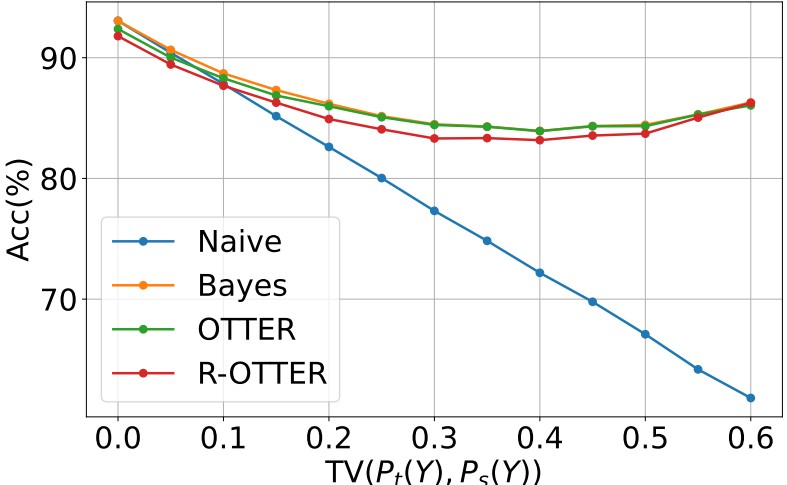

Figure 8: Synthetic experiment result with R-OTTER. As expected, R-OTTER can successfully resolve the effects of label shift.

- fish: aquarium fish, flatfish, ray, shark , trout

- tree: maple tree, oak tree, palm tree, pine tree, willow tree

**Class balance estimation error**   We report the class balance estimation error in Section 5.4. Table 14 shows the total variation between true class balance and estimated class balance. We can expect a significant accuracy improvement for RN50, RN101, and ViT-B/16 based on this table.

### E.6   Synthetic experiments with R-OTTER

We validate our theoretical result (Theorem 4.4) by testing R-OTTER in a fully controllable setting.

**Setup and Procedure.**   We use our synthetic experiment setup (Section 5.2) with perturbation noise $\delta = 0$ and label distribution $\alpha = 0$. Additionally, we generated a validation set that follows the same distribution as the test set. After learning the reweighting parameter $r$ in the validation set using $y_{otter}$ as pseudolabels, we evaluated R-OTTER on the test set, comparing the results to those of zero-shot and OTTER. We expect that, if successful, R-OTTER will similarly gain improvement over zero-shot when the source and target distributions increasingly differ.

**Results**   Figure 8 presents the experimental result. As expected, R-OTTER presents the experimental results. As expected, R-OTTER performs closely to the Bayes optimal solution, demonstrating its effectiveness, though it exhibits slight suboptimality due to generalization issues.

