# OpenReview forum: "OTTER: Effortless Label Distribution Adaptation of Zero-shot Models"
_NeurIPS.cc/2024/Conference — NeurIPS 2024 poster_

### Official Review · Reviewer_t22L · 2024-07-01

**Soundness:** 4
**Presentation:** 3
**Contribution:** 3
**Rating:** 7
**Confidence:** 4

**Summary:**

This paper introduces a novel approach to address the issue of label distribution mismatch in zero-shot models, which is a common problem due to the imbalance in the pretraining datasets. The proposed method uses optimal transport to adjust the predictions of pretrained models based on an estimated downstream label distribution, without requiring additional training or access to labeled downstream data.

**Strengths:**

1. The use of optimal transport to handle label distribution mismatch in zero-shot models is a novel and elegant solution.
2. The paper provides a solid theoretical foundation for the proposed method, including characterizations of the improvement under mild conditions and error bounds for misspecification.
3. Extensive empirical validation across a wide range of image and text classification tasks demonstrates the effectiveness of OTTER.

**Weaknesses:**

1. dependence on estimated label distribution: the method relies on an accurate estimate of the downstream label distribution. In practice, obtaining a reliable estimate may be challenging, and errors in this estimate could impact the performance of OTTER.
2. evaluation on diverse datasets: while the paper includes a variety of datasets for validation, a more detailed analysis of performance across different types of datasets (e.g., varying in size, complexity, and imbalance levels) would provide a deeper understanding of the method's strengths and limitations.
3. comparison with recent methods: the paper compares OTTER with some existing baselines, but it could benefit from a more comprehensive comparison with the latest state-of-the-art methods in label distribution adaptation and zero-shot learning. There are many training-free or test-time adaption zero-shot baselines[a][b][c]. How does OTTER perform compared to these methods?
4. scalability: although described as lightweight, the paper does not thoroughly discuss the scalability of OTTER for very large-scale datasets.


[a] Mirza, Muhammad Jehanzeb, et al. "Lafter: Label-free tuning of zero-shot classifier using language and unlabeled image collections." NIPS 2024.

[b] Zhao, Shuai, et al. "Test-time adaptation with clip reward for zero-shot generalization in vision-language models." arXiv preprint arXiv:2305.18010.

[c] Abdul Samadh, Jameel, et al. "Align your prompts: Test-time prompting with distribution alignment for zero-shot generalization." NIPS 2024.

**Questions:**

The paper presents a compelling and well-validated approach to address label distribution mismatch in zero-shot models using optimal transport. Its innovative method, strong theoretical foundation, and significant empirical improvements make it a valuable contribution to the field. However, further exploration of its limitations, more comparisons with recent methods, and practical implementation details would enhance its impact and utility.

**Limitations:**

The authors do not claim limitations.

---

> ### Author Rebuttal · Authors · 2024-08-07
>
> We are grateful to the reviewer for noting the strengths of our work and providing useful comments. The reviewer appreciated our work, recognizing the novelty of our method, its solid theoretical foundation, and extensive empirical validation.
>
> * **On dependence on estimated label distribution**: While our method requires estimated label distribution, the requirement for improvement is not heavy---the estimated distribution just needs to be better than zero-shot label distribubtion. Furthermore, label distribution estimation algorithms have been developed in other  works (e.g. [1,2,3,4,5]). Any availble method can be plugged into OTTER. For example, we used BBSE [1] as a label distribution estimation method in linear probing setting.
>
> * **On more detailed analysis of performance across different types of datasets**: We reported information about datasets in Appendix E.1 Table 6, which shows that experiments cover diverse datasets with respect to the number of data points, the number of classes and the imbalance level. Additionally, we provide further analysis illustrating the relationship between (n, K, Imbalance) and OTTER's performance gains in Figure 3 of the attached pdf. In summary, **accuracy tends to increase as dataset size increases, the number of classes decreases, and imbalance level increases**.
>
> * **On comparison with recent methods**: Thank you for suggesting useful references. The suggested papers share commonalities in seeking to perform test-time adapation without any labeled data. [6] fine-tunes zero-shot models using pseudo-labels generated by text classifiers trained on LLM-generated text descriptions and class names. [7] uses CLIP scores as reward signals in reinforcement learning, facilitating task-specific fine-tuning of VLMs. [8] uses test-time prompt tuning with alignment loss to effectively align representations with pretraining data. While OTTER is indeed also a test-time alignment method, **it offers a much more lightweight way to update predictions without any parameter updates, while [6, 7, 8] require additional training for some parts of models**.
> Additionally, OTTER can be easily combined with [6, 7, 8], giving further improvement. We include a mini experiment with [6] in three datasets: Caltech101, DTD, OxfordFlowers.
>
> |                | Caltech101 | DTD   |  OxfordFlowers |
> | -------------- | ---------- | ----- |  ------------- |
> | Zeroshot       | 90.67      | 42.02 |  63.78         |
> | LaFTer         | 93.06      | 49.05 |  71.70         |
> | OTTER          | 94.73      | 45.98 |  68.01         |
> | LaFTer + OTTER | **95.90**      | **53.72** |  **76.29**         |
>
> OTTER is comparable to LaFTer (but, as we described, more lightweight) and  provides further improvements when combined with LaFTer.
>
> * **On scalability**: As mentioned in the common response, we reported computation time in Appendix E.3. Table 10. While it is true that our inference-time adaptation approach requires additional computation, **the computational overhead is not heavy** --- the linear programming version of OT can run with subqudratic computation comlexity. In practice, we observed our method gives modified predictions **within 0.05 ms per sample**---a negligible overhead
> Additionally, for massive-scale inference, batch optimal transport with parallel computing can be used. Figure 1 in the attached file shows the accuracy and computation time (per batch) depending on the batch size. Note that this result can be further improved with more advanced batch optimal transport methods.
>
> [1] Saerens, Marco, Patrice Latinne, and Christine Decaestecker. "Adjusting the outputs of a classifier to new a priori probabilities: a simple procedure." Neural computation 14.1 (2002): 21-41.
>
> [2] Lipton, Zachary, Yu-Xiang Wang, and Alexander Smola. "Detecting and correcting for label shift with black box predictors." ICML'18.
>
> [3] Azizzadenesheli, Kamyar, et al. "Regularized Learning for Domain Adaptation under Label Shifts." ICLR'18.
>
> [4] Alexandari, Amr, Anshul Kundaje, and Avanti Shrikumar. "Maximum likelihood with bias-corrected calibration is hard-to-beat at label shift adaptation." ICML'20.
>
> [5] Garg, Saurabh, et al. "A unified view of label shift estimation." NeurIPS'20.
>
> [6] Mirza, Muhammad Jehanzeb, et al. "Lafter: Label-free tuning of zero-shot classifier using language and unlabeled image collections." NeurIPS'24.
>
> [7] Zhao, Shuai, et al. "Test-time adaptation with clip reward for zero-shot generalization in vision-language models." arXiv preprint arXiv:2305.18010.
>
> [8] Abdul Samadh, Jameel, et al. "Align your prompts: Test-time prompting with distribution alignment for zero-shot generalization." NeurIPS'24.

---

> > ### Comment · Reviewer_t22L · 2024-08-10
> >
> > Thanks for the authors' answer. Regarding Q2, can you explain why the accuracy increases with the increase of imbalance? This is counterintuitive.

---

> > > ### Author Response · Authors · 2024-08-10
> > >
> > > Thank you for your follow-up question. The reason OTTER shows greater accuracy gains with increased class imbalance is that there's more room for improvement in these scenarios. When the downstream class distribution is imbalanced, it often deviates more significantly from the concept distribution in the pretraining data, leading to a larger label distribution shift. This greater discrepancy allows OTTER to make more impactful corrections, resulting in the observed accuracy increase.

---

### Official Review · Reviewer_M3dD · 2024-07-02

**Soundness:** 3
**Presentation:** 3
**Contribution:** 1
**Rating:** 5
**Confidence:** 4

**Summary:**

Zero shot classification suffers from label distribution mismatch. This paper suggest to adjust pretrained model prediction via optimal transport. By showing the optimal transport prediction is equivalent to the bayes optimal classifier's output theoretically and good model performances on various experimental settings empirically, the paper shows the validity of its proposed method.

**Strengths:**

- The problem is well formulated
- Various experiment

**Weaknesses:**

- Novelty is limited. This paper suggests applying OT for managing label distribution shift. The class imbalance for pretrained network prediction is already well known, and OT has been already applied for various distribution shift cases. Therefore, naively applying OT for the biased pretrained model prediction is imaginable.
- The proposal saying that Bayes optimal classifier can be derived through optimal transport is inadequate and too strong since it requires a true cost matrix. If $P_t(Y|X)$ is accessible, we don't even need to use the optimal transport.
- Please link theorems of the main paper and proofs in the appendices for readability.
- I don't think figure 2 shows OTTER remains unaffected. The acc is deteriorated from 90 to 7X, meaning about 15%p decrease.

**Questions:**

- Considering each cell of the cost matrix $C$, it is a loss (or the classifier score) calculated from the pretrained network. It means the cost matrix is biased to the source data distribution. In that sense, how the optimal transport can be considered as assigning true label?
- How can the true label distribution of the target, $\nu*$, be accessible?
- Can the author show more details for the equation of section 4.1? I cannot understand how to remove $P_s(X)$ and $P_t(X)$
- In table 1, what happened to Caltech256 Prior matching? Why the performance is far worse than zero-shot?
- In figure 2, Why OTTER shows acc increase when the total variation distance between the source and the target increases from 0.4 to 0.8? With larger distribution shift, acc is supposed to deteriorate. If not, the synthetic experiment may be poorly modeled.
- How long does it take to apply OTTER? I suppose it will take quadratically proportional to the dataset size.

**Limitations:**

.

---

> ### Author Rebuttal · Authors · 2024-08-07
>
> We appreciate the thoughtful comments and references. We will include links to proofs and will clarify the statements corresponding to the reviewer's questions in our updated draft.
>
> * **On the proposal saying that Bayes optimal classifier can be derived through optimal transport**: The statement in L151 means OTTER does not cause harm when the cost matrix has the true target posterior $P_t(Y|X)$; we do not need $P_t(Y|X)$ to make OTTER work well. This result is connected to Theorem 4.2 in Section 4.1. Since the assumption in Theorem 4.2 makes OTTER with $C_{ij}=P_{\theta}(Y|X)$ and OTTER with $C_{ij}^*=P_{t}(Y|X)$ equivalent, OTTER can achieve the Bayes optimal classifier performance. We will make this connection clear.
>
> * **On the interpretation of Figure 2**: In Figure 2(a), $\delta$ represents noise in prediction scores, which affects calibration. Such errors may affect the performance of OTTER. *Without perturbation ($\delta=0$), OTTER remains unaffected by the label distribution shift* (L265-266), achieving a Bayes optimal classifier.
>
> Additional questions:
> * *Considering each cell ... assigning true label?*: Our theorem shows that even when the cost matrix is biased to the source data distribution, optimal transport can fix the bias from the source data distribution using the label distribution specification. It does not necessarily assign true labels---it yields better predictions when the label distribution specification is better than the label distribution of biased zero-shot predictions, given proper calibration. We validate this finding by observing the performance gain in our experiments.
> * *How can the true label distribution of the target, $\nu^*$, be accessible?*: **This is not necessary**---see our common response and additional experiments. We also note that there are many label distribution estimation algorithms (e.g. [1,2,3,4,5]). **Any available method can be plugged into OTTER**. For example, we used BBSE [1] as a label distribution estimation method in the linear probing setting.
>
> * *More details for the equation of section 4.1?*: On Equation in Section 4.1. Equation in Section 4.1. mainly proceeds with Bayes Rule and invariance assumption $P_s(X|Y)$=$P_t(X|Y)$.
>
> $\tilde{s}_{\theta}$
>
> $=s_{\theta}\frac{P_t(Y=j)}{P_s(Y=j)} \qquad \because$ by adjustments
>
> $=P_s(Y=j|X=x)\frac{P_t(Y=j)}{P_s(Y=j)} \qquad \because$ by the assumption $s_{\theta}=P_s(Y=j|X=x)$
>
> $=\frac{P_s(X=x|Y=j)P_s(Y=j)}{P_s(X=x)}\frac{P_t(Y=j)}{P_s(Y=j)} \quad \because$ by Bayes rule
>
> $=\frac{P_s(X=x|Y=j)P_t(Y=j)}{P_s(X=x)}$
>
> $=\frac{P_t(X=x|Y=j)P_t(Y=j)}{P_s(X=x)} \quad \because$ by the assumption $P_s(X=x|Y=j)=P_t(X=x|Y=j)$
>
> $\propto P_t(X=x|Y=j)P_t(Y=j)$
>
> $=\frac{P_t(Y=j|X=j)P_t(X=x)}{P_t(Y=j)}P_t(Y=j) \quad \because$ by Bayes rule
>
> $=P_t(Y=j|X=j)P_t(X=x)$
>
> $\propto P_t(Y=j|X=j)$
>
> Here, $P_s(X=x)$ and $P_t(X=x)$ can be omitted using $\propto$ as they do not affect the classification output.
>
> * *In table 1, what happened to Caltech256 Prior matching?*: We found that prior matching is sensitive to the learning rate and temperature hyperparameters---but hyperparameter tuning with plentiful data is typically not an option for zero-shot models. In the experiments for Table 1, we selected hyperparameters of prior matching via grid search, by evaluating their performance on small validation data (10 shots per class). Prior matching's accuracy in Caltech256 shows the case hyperparameter search fails, which leads to suboptimal solutions in the optimization of the weights $r$ (L557-558).
>
> * *In figure 2, Why OTTER shows acc increase...*: The Bayes error rate changes with the label shift. Thus accuracy of Bayes optimal classifier changes, and OTTER follows the accuracy of Bayes optimal classifier. Specifically, given $\mathcal{N}(-1,1)$ and $X|Y=1 \sim \mathcal{N}(1,1)$, the error rate $\int_x1-\max_yP(Y=y|X=x)$ is maximized at $\nu^s_0$=$\nu^s_1$=0.5. Total variation 0.4 corresponds to this point, and the Bayes error rate is reduced after that. OTTER achieves the Bayes error regardless of label distribution shift, while naive classification deteriorates with the magnitude of the label distribution shift.
>
> * *How long does it take to apply OTTER? I suppose it will take quadratically proportional to the dataset size.*: We report computation time in Appendix E.3. Table 10. The computational complexity  depends on implementation. The linear programming version of the optimal transport algorithm can run in $\tilde{O}(nk\sqrt{n+k})$ time via minimum cost flow [6], where $n$ is the number of data points and $k$ is the number of classes. Thus, computation time **subquadratically** increases in the number of data points. In practice, we observed our method gives modified predictions **within 0.05 ms per sample**---a negligible overhead. Additionally, a batched version with parallel computing is an option for massive inference. We included an experiment where the performance of optimal transport changes depending on batch size in Figure 1 of the attached pdf. The result shows that reasonable accuracy improvements can be obtained even when randomly partitioning data and applying OTTER.
>
> We appreciate the reviewer's consideration and are more than willing to address any further concerns. If we have adequately resolved the issues, we would be grateful if the reviewer could consider raising their score.

---

> ### Author Response · Authors · 2024-08-07
> **Reference**
>
> [1] Saerens, Marco, Patrice Latinne, and Christine Decaestecker. "Adjusting the outputs of a classifier to new a priori probabilities: a simple procedure." Neural computation 14.1 (2002): 21-41.
>
> [2] Lipton, Zachary, Yu-Xiang Wang, and Alexander Smola. "Detecting and correcting for label shift with black box predictors." ICML'18.
>
> [3] Azizzadenesheli, Kamyar, et al. "Regularized Learning for Domain Adaptation under Label Shifts." ICLR'18.
>
> [4] Alexandari, Amr, Anshul Kundaje, and Avanti Shrikumar. "Maximum likelihood with bias-corrected calibration is hard-to-beat at label shift adaptation." ICML'20.
>
> [5] Garg, Saurabh, et al. "A unified view of label shift estimation." NeurIPS'20.
>
> [6] Lee, Yin Tat, and Aaron Sidford. "Path finding methods for linear programming: Solving linear programs in $\tilde{O}(\sqrt{rank})$ iterations and faster algorithms for maximum flow." IEEE 55th Annual Symposium on Foundations of Computer Science. IEEE, 2014.

---

> ### Comment · Area_Chair_66Hy · 2024-08-12
> **Respond to author rebuttal**
>
> Dear reviewer,
>
> Thank you for your assistance in the review process. Could you please read the author rebuttal ASAP and let them / fellow reviewers know if it addressed your concerns?

---

> > ### Comment · Reviewer_M3dD · 2024-08-13
> >
> > I checked the authors' rebuttal, and appreciate their effort to relieve my concerns.
> >
> > Since my questions for the novelty claim has not been solved, and the authors have thoroughly answered my questions, I increase my score to 5.

---

### Official Review · Reviewer_t3Bz · 2024-07-08

**Soundness:** 3
**Presentation:** 3
**Contribution:** 3
**Rating:** 8
**Confidence:** 4

**Summary:**

Zero-shot models such as CLIP suffer from imbalanced predictions inherited from the uncurated pre-training data. This paper proposes adjusting the predictions of zero-shot models via optimal transport. This method requires only the downstream data and the label prior of the target distribution. Under the assumption that the conditional likelihood is unchanged between the pre-training data and the downstream data, the authors prove that the induced predictions are Bayes Optimal. The authors validate the effectiveness of the proposed method on several benchmark tests.

**Strengths:**

The paper is well-organized and the presentation is clear. The idea of using optimal transport to adjust zero-shot predictions on downstream data is novel to me. Besides, the authors prove that the resulting predictions are Bayes optimal and justify the cost matrix using posteriors of zero-shot models. I have coarsely checked the proofs and find them convincing.

**Weaknesses:**

Major:

(1) Related works are missing. Some papers have addresses the issue of removing the label bias in zero-shot models. For instance, [1][2] use a subset of the pre-training data to mitigate the bias. [3] focuses on only using the labeled downstream data to address imbalanced predictions and proposes two methods. The authors should consider including these prior works.

[1] A simple zero-shot prompt weighting technique to improve prompt ensembling in text-image models.

[2] The neglected tails of vision-language models.

[3] Generalized Logit Adjustment: Improved Fine-tuning by Mitigating Label Bias in Foundation Models.

(2) The proposed method is transductive and requires the entire test dataset to perform optimal transport. I am wondering if OTTER can estimate the label distribution of pre-training data $\pi$ from unlabeled downstream training data and then the debiasing procedure can be performed by subtracting the logarithm $\ln \pi$ from the zero-shot predictions, as described in [3] (https://arxiv.org/pdf/2310.08106), without needing to operate on the entire test dataset

Minor:

(1) Duplicate citations: [29] and [30].

(2) Line 120: should reference Algorithm 1.

**Questions:**

Please refer to the weaknesses, and I am willing to raise the score if the major weaknesses are addressed.

Additionally, I have a question to discuss with the authors. Both [3] and OTTER use downstream data to generate Bayes optimal predictions based on the assumption that $P(x|y)$ remains unchanged between the pre-training distribution and the downstream distribution. In practice, this assumption might not hold. For instance, the pre-training data might mostly consist of real photos, while some test benchmarks might be sketches, e.g., ImageNet-Sketch. Under this situation, is your method theoretically sound in mitigating the pre-training bias?

**Limitations:**

Limitations have been discussed and no negative societal impact has been identified.

---

> ### Author Rebuttal · Authors · 2024-08-07
>
> We sincerely thank the reviewer for their kind words, constructive feedback, and useful suggestions. We appreciate the reviewer recognizing the novelty of our work and its strong theoretical foundation. We plan to include the suggested related works and fix typos.
>
> * **On additional related works (Major 1)**: Thank you for providing more related works. We plan to include the suggested papers in our related works with the following discussion.
> [1, 2, 3] address the bias induced by the concept distribution in pretraining data, in a similar context as OTTER.
> [1] tackles word frequency bias in pretraining and spurious concept frequency bias in test data in prompt ensembles. This work normalizes/debiases logit scores with the logits from the pretraining data to reweight/select prompts.
> [2] addresses concept frequency bias in pretraining data as well via retrieval augmented prompting, i.e. retrieving the most frequent concept from synonyms of the label. First, they propose a new concept frequency estimation method with LLMs. By concept frequency estimation, they re-confirm long-tailed concept distribution, and they propose retrieval-based prompting/linear classifier training.
> Similarly, [3] adjusts logits to debias pretraining / fine-tuning label distribution. They propose a pretraining label distribution estimation method with an optimization formulation and show that the proposed adjustment make the finetuning.
> The main distinction of OTTER from these works is that it does not require access to the pretraining data distribution, as long as the label distribution specification is properly given. This is particularly advantageous because pretraining data is often inaccessible due to proprietary or privacy issues.
>
> * **On the comparison with [3] (Major 2)**: [3] tackles a similar problem in the sense they try to mitigate bias from pretraining, but **there are several key differences**: 1) [3] mainly considers fine-tuned models and uses ensembles of these fine-tuned models and zero-shot models, while our work **focuses exclusively on zero-shot models**. 2) [3] assumes a uniform label distribution in target distribution for their Bayes classifier result. If we make such an assumption, OTTER can achieve the Bayes classifier performance **without** access to the pretraining distribution. [3] presented an extension for label shifts in the target distribution in its appendix, which requires the target label distribution as well. Also, while our method often uses the entire test set, batch optimal transport can be applied in the case the large-scale inference is required (as we described in the common response). In practice, applying OTTER to inference set with $n<20000$ usually takes less than 1 second, as can be seen in Appendix E.3. Table 10. We added additional experimental results with the batched version of OTTER (randomly partition test data and apply OTTER with the global label distribution specification) in Figure 2 (attached pdf). The **results show that batched version still improves accuracy, enabling parallel processing**. We expect batched OTTER can be further improved with more recent works on batched optimal transport [4, 5].
>
> * **On the label shift assumption that P(X|Y) does not change**: If P(X|Y) changes, the invariance result does not hold (it is the same for [3]). Instead, our error bound in Section 4.3. provides an accuracy bound. Suppose the posterior ratio is given by $\epsilon_{ij}=\frac{P_t(X=x_i|Y=j)}{P_s(X=x_i|Y=j)}$ in the setup of Appendix D.2. Proof of Theorem 4.2. Then, the proof lines change as follows
>
> $C^*_{ij}$
>
> $= - \log P_t(Y=j|X=x_i)$
>
> $= - \log \frac{P_t(X=x_i|Y=j)P_t(Y=j)}{P_t(X=x_i)}$
>
> $= - \log \frac{\epsilon_{ij}P_s(X=x_i|Y=j)P_t(Y=j)}{P_t(X=x_i)}$
>
> $= - \log \frac{\epsilon_{ij}P_s(Y_j|X=x_i)P_s(X=x_i)P_t(Y=j)}{P_s(Y=j)P_t(X=x_i)}$
>
> $= -\log\epsilon_{ij}-\log P_s(Y=j|X=x_i)\frac{P_s(X=x_i)P_t(Y=j)}{P_t(X=x_i)P_s(Y=j)} -\log P_s(Y=j|X=x_i) + \log P_s(Y=j)$
>
> $= -\log\epsilon_{ij}+C_{ij} + E_{\cdot j} + F_{i \cdot}$
>
> This shows that when the label shift assumption is violated, OTTER with $C_{ij}=-\log P_{\theta}(Y=j|X=x_i)$ yields equivalent predictions with $C_{ij}^* + \log{\epsilon_{ij}}$ rather than $C^*_{ij}$, making it suboptimal. The similar deviation can be observed in the Proof of Proosition of [3] by plugging $\epsilon_{yz}=\frac{P_t(z|y)}{P_p(z|y)}$ into Equation (28), (29).
> However, in our analysis, Theorem 4.3. shows the additional error rate linearly depends on the log ratio of $P_t(X|Y)$ and $P_s(X|Y)$ by defining cost matrix gap as $\Delta_{C_{ij}} = \log \epsilon_{ij}$. Thus, if the deviation is not significant, **we can expect near-invariant results. In practice, as we can see in ImageNet-Sketch results, OTTER works well even when P(X|Y) invariance is violated**.
>
> We appreciate the reviewer's consideration and are more than willing to address any further concerns. If we have adequately resolved the issues, we would be grateful if the reviewer could consider raising their score.
>
> [1] Allingham, James Urquhart, et al. "A simple zero-shot prompt weighting technique to improve prompt ensembling in text-image models." ICML'23.
>
> [2] Parashar, Shubham, et al. "The Neglected Tails in Vision-Language Models." CVPR'24.
>
> [3] Zhu, Beier, et al. "Generalized logit adjustment: Calibrating fine-tuned models by removing label bias in foundation models." NeurIPS'23
>
> [4] Nguyen, Khai, et al. "Improving mini-batch optimal transport via partial transportation." ICML'22.
>
> [5] Nguyen, Khai, et al. "On Transportation of Mini-batches: A Hierarchical Approach." ICML'22.

---

> ### Comment · Reviewer_t3Bz · 2024-08-08
>
> Thank you for your response. Regarding your reply to W2, if I understand correctly, the authors suggest that the proposed OTTER method at least requires a batch of test samples to produce an unbiased prediction. My question is, if we are given a set of i.i.d. validation data with zero-shot predictions $y_{zs}$ and debiased predictions $y_{otter}$, is it possible to use these terms to derive $(\log\frac{P_t(y)}{P_s(y)}$such that the requirement of using all test data or a batch of test samples can be avoided?

---

> > ### Author Response · Authors · 2024-08-11
> >
> > Thank you for the interesting idea! Yes, combining two methods in the way you suggest could eliminate the need for the entire dataset or large data subsets in optimal transport. We appreciate the suggestion and will add the combination to our draft.

---

> > > ### Comment · Reviewer_t3Bz · 2024-08-11
> > >
> > > Thank you for your updates. In my opinion, combining OTTER with [3] to avoid transductive zero-shot predictions would make the proposed method more practical. As a result, I am strongly considering increasing my score for the paper if the revised version incorporates this suggested combination. If you agree with the suggestion, could you provide concrete details on what changes you plan to make and where they will be implemented?

---

> > > > ### Author Response · Authors · 2024-08-12
> > > >
> > > > Thank you! We appreciate it.
> > > >
> > > > Following your suggestion, we have updated our paper to introduce a variant of our approach called R-OTTER (Reweighting OTTER). We added the R-OTTER formulation to Section 3, a set of theoretical results (which we sketch here) to Section 4, and empirical results (highlighted below, including both synthetic and real data settings) to Section 5. This approach helps reduce the reliance on large data subsets or the entire dataset for prediction.
> > > >
> > > > R-OTTER involves learning a reweighting factor (an estimate of $P_t(Y)/P_s(Y)$) using OTTER's predictions $y_{otter}$ as pseudolabels in validation set. This reweighting can then be applied to adjust probability scores or logits without relying on the entire test dataset.
> > > >
> > > > * R-OTTER formulation: For simplicity, we use a reweighting formulation equivalent to logit adjustment in [3]. The reweighted probability scores of $P_\theta$ with a reweighting parameter vector $r \in R^{K}$ defined as $P_{\theta, r}(Y=j|X=x) = \cfrac{r_j P_\theta(Y=j|X=x)}{\sum_{j'=1}^K r_{j'} P_\theta(Y=j'|X=x)}$. We learn $r$ with $y_{otter}$ using a cross entropy loss.
> > > >
> > > > We expect R-OTTER to perform comparably to OTTER (but with the desired benefit of not requiring running OTTER over the entire dataset) by learning the reweighting factor accurately. We provide a theoretical result showing that $r^*=P_t(Y)/P_s(Y)$ is optimal in this formulation.
> > > >
> > > > * **Theorem**. Under the same assumption of Theorem 4.2., $r^*=P_t(Y)/P_s(Y)$ is an optimal parameter when learning with $y_{otter}$.
> > > >
> > > > **Proof (Sketch)**. By assumption $P_\theta(Y|X)=P_s(Y|X)$. Using the result of Theorem 4.2, $y_{otter}$ samples are generated by
> > > > $\arg\max_{j}P_t(Y=j|X=x)$. Suppose $r^*=P_t(Y)/P_s(Y)$. Then,
> > > >
> > > > $P_{\theta, r^*}(Y=j|X=x)$
> > > >
> > > > $= \cfrac{r_j^* P_\theta(Y=j|X=x)}{\sum_{j'=1}^K r^*_{j'} P_\theta(Y=j'|X=x)}$
> > > >
> > > > $=\cfrac{r_j^* P_s(Y=j|X=x)}{\sum_{j'=1}^K r_{j'}^* P_s(Y=j'|X=x)}$
> > > >
> > > > $=\cfrac{\frac{P_t(Y=j)}{P_s(Y=j)}\frac{P_s(X=x|Y=j)P_s(Y=j)}{P_s(X=x)}}{\sum_{j'=1}^K\frac{P_t(Y=j')}{P_s(Y=j')}\frac{P_s(X=x|Y=j')P_s(Y=j')}{P_s(X=x)}}$
> > > >
> > > > $=\cfrac{P_t(Y=j)P_s(X=x|Y=j)}{\sum_{j'=1}^KP_t(Y=j')P_s(X=x|Y=j')}$
> > > >
> > > > $=\cfrac{P_t(Y=j)P_t(X=x|Y=j)}{\sum_{j'=1}^KP_t(Y=j')P_t(X=x|Y=j')}$
> > > >
> > > > $=\cfrac{\frac{P_t(Y=j)P_t(Y=j|X=x)P_t(X=x)}{P_t(Y=j)}}{\sum_{j'=1}^K\frac{P_t(Y=j')P_t(Y=j'|X=x)P_t(X=x)}{P_t(Y=j')}}$
> > > >
> > > > $=\cfrac{P_t(Y=j|X=x)}{\sum_{j'=1}^K P_t(Y=j'|X=x)}$
> > > >
> > > > Thus, $y_{r\text{-}otter} = \arg\max_{j}P_{\theta, r^*}(Y=j|X=x)$
> > > >
> > > > $=\arg\max_{j}\cfrac{P_t(Y=j|X=x)}{\sum_{j'=1}^K P_t(Y=j'|X=x)}$
> > > >
> > > > $=\arg\max_{j}P_t(Y=j|X=x)$
> > > >
> > > > $=y_{otter}$.
> > > >
> > > > Our synthetic experiment validates the above theorem.
> > > > * **Synthetic experiment results**: We use our synthetic experiment setup with perturbation noise $\delta=0$ and label distribution $\alpha=0$. Additionally, we generated a validation set that follows the same distribution as the test set. After learning the reweighting parameter $r$ in the validation set using $y_{otter}$ as pseudolabels, we evaluated R-OTTER on the test set, comparing the results to those of zero-shot and OTTER. We expect that, if successful, R-OTTER will similarly gain improvement over zero-shot when the source and target distributions increasingly differ. We observe this:
> > > >
> > > > | TV (P_t(Y), P_s(Y)) | ZS     | OTTER  | R-OTTER | Bayes Optimal |
> > > > |-----------------|--------|--------|---------|---------------|
> > > > | 0               | 0.9306 | 0.9238 | **0.9179**  | 0.9305        |
> > > > | 0.05            | 0.9041 | 0.9001 | **0.8944**  | 0.9065        |
> > > > | 0.1             | 0.8783 | 0.8830 | **0.8768**  | 0.8870        |
> > > > | 0.15            | 0.8516 | 0.8687 | **0.8628**  | 0.8732        |
> > > > | 0.2             | 0.8261 | 0.8598 | **0.8492**  | 0.8618        |
> > > > | 0.25            | 0.8003 | 0.8507 | **0.8407**  | 0.8516        |
> > > > | 0.3             | 0.7731 | 0.8443 | **0.8331**  | 0.8449        |
> > > > | 0.35            | 0.7483 | 0.8428 | **0.8334**  | 0.8430        |
> > > > | 0.4             | 0.7218 | 0.8392 | **0.8317**  | 0.8393        |
> > > > | 0.45            | 0.6978 | 0.8432 | **0.8355**  | 0.8434        |
> > > > | 0.5             | 0.6709 | 0.8433 | **0.8371**  | 0.8444        |
> > > >
> > > > Overall, we observe comparable accuracy to OTTER, where the difference is controlled by the parameter estimation error.
> > > >
> > > > We anticipate similar findings in our real-world experiments:
> > > >
> > > > * **Real experiment results**: We conducted a mini real-world experiment by splitting the original dataset into validation and test sets, following the same procedure as in the synthetic experiment.
> > > >
> > > > |            | ZS     | OTTER  | R-OTTER |
> > > > |------------|--------|--------|---------|
> > > > | CIFAR100   | 0.6381 | 0.6759 | **0.6532**  |
> > > > | Caltech101 | 0.7964 | 0.8541 | **0.8427**  |
> > > > | DTD        | 0.3893 | 0.4381 | **0.4156**  |
> > > > | EUROSAT    | 0.3286 | 0.4185 | **0.3453**  |
> > > > | Flowers102 | 0.6382 | 0.6894 | **0.6836**  |
> > > >
> > > > As expected, R-OTTER, indeed performs better than zero-shot across all datasets and often comparably to OTTER. We will include the full results in the revised manuscript.

---

> > > > > ### Comment · Reviewer_t3Bz · 2024-08-12
> > > > >
> > > > > The theory and experiments sound good to me. I am happy to increase my score to 8.

---

### Author Rebuttal · Authors · 2024-08-07

### Common Response
We thank all of the reviewers for their kind comments and feedback. Reviewers recognized the strengths of our paper:
* OTTER provides **a novel and elegant solution** to deal with label distribution using optimal transport. (Reviewers t3Bz, t22L)
* OTTER offers **theoretical results** showing that (1) OTTER can recover a Bayes optimal classifier under the label shift setup, and (2) the error bound can be derived with misspecification of label distribution estimation and miscalibration. (Reviewers t3Bz, t22L)
* **Empirical results** in a variety of experimental settings demonstrate the effectiveness of OTTER. (Reviewers M3dD, t22L)

We address two common questions before proceeding to individual responses:
* **On computation cost and scalability of OTTER**: We report computation time in Appendix E.3. Table 10. While it is true that our inference-time adaptation approach requires additional computation, **the computational overhead is not heavy**. The linear programming version of the optimal transport algorithm can run in $\tilde{O}(nk\sqrt{n+k})$ time via minimum cost flow [1], where $n$ is the number of data points and $k$ is the number of classes. Thus, computation time subquadratically increases in the number of data points. In practice, we observed our method gives modified predictions **within 0.05 ms per sample**---a negligible overhead.
Additionally, batched OTTER with parallel computing, instead of using the full inference dataset, can be used for massive-scale inference. Figure 1 in the attached file shows the accuracy and computation time (per batch) depending on the batch size. Note that this result can be further improved with more advanced batched optimal transport methods [2, 3].

* **On the dependency on estimated label distribution**: While the true label distribution enables the maximum improvement when using the proposed method, **it is not necessary**. Indeed, our algorithm can improve zero-shot classification *with just slightly better label distribution estimation than the one implicitly used in zero-shot prediction*---so that our requirements are extremely low. To illustrate this claim, we interpolate the label distribution of zero-shot prediction $\nu^{zs}$ and the true label distribution $\nu^{true}$ such that $$\hat{\nu}_\alpha=(1-\alpha)\nu^{zs} + \alpha\nu^{true},$$ where $0 \leq \alpha \leq 1$. We use $\hat{\nu}_\alpha$ as the label distribution specification for OTTER and provide a graph (Figure 2 in the attached pdf) that illustrates the resulting accuracy changes. As expected, **as long as the specification of the label distribution is closer to the true distribution, our technique shows performance improvement in all cases**.

[1] Lee, Yin Tat, and Aaron Sidford. "Path finding methods for linear programming: Solving linear programs in $\tilde{O}(\sqrt{rank})$ iterations and faster algorithms for maximum flow." 2014 IEEE 55th Annual Symposium on Foundations of Computer Science. IEEE, 2014.

[2] Nguyen, Khai, et al. "Improving mini-batch optimal transport via partial transportation." ICML'22.

[3] Nguyen, Khai, et al. "On Transportation of Mini-batches: A Hierarchical Approach." ICML'22.

---

### Decision · Program_Chairs · 2024-09-25

**Decision:**

Accept (poster)

**Comment:**

The paper provides a simple method to handle changes in label proportions between a pretrained model and a downstream task, via optimal transport. The method is simple and straightforward, and the authors perform a comprehensive empirical evaluation backed by theory. The paper is generally well-written and easy to read. Thereby, I recommend acceptance as a poster.